# Agent World Model:
# Infinity Synthetic Environments for Agentic Reinforcement Learning

**Zhaoyang Wang** [1]   **Canwen Xu** [2]   **Boyi Liu** [2]   **Yite Wang** [2]   **Siwei Han** [1]
**Zhewei Yao** [2]   **Huaxiu Yao** [*1]   **Yuxiong He** [*2]

## Abstract

Recent advances in large language model (LLM) have empowered autonomous agents to perform multi-turn interactions with tools and environments. However, scaling such agent training is limited by the lack of diverse and reliable environments. In this paper, we propose **A**gent **W**orld **M**odel (**AWM**), a fully synthetic environment generation pipeline. Using this pipeline, we scale to 1,000 environments covering everyday scenarios, in which agents can interact with rich toolsets and obtain high-quality observations. Notably, these environments are code-driven and backed by databases, providing more reliable and consistent state transitions than environments simulated by LLMs. Moreover, they enable more efficient agent interaction compared with collecting trajectories from realistic environments. To demonstrate the effectiveness of this resource, we perform large-scale reinforcement learning for multi-turn tool-use agents. Thanks to the fully executable environments and accessible database states, we can also design reliable reward functions. Experiments on three benchmarks show that training exclusively in synthetic environments, rather than benchmark-specific ones, yields strong out-of-distribution generalization. The code is available at `https://github.com/Snowflake-Labs/agent-world-model`.

## 1. Introduction

Large language models (LLMs) have achieved remarkable performance in instruction following, reasoning, code generation, and tool-use (Chen et al., 2021; Schick et al., 2023;

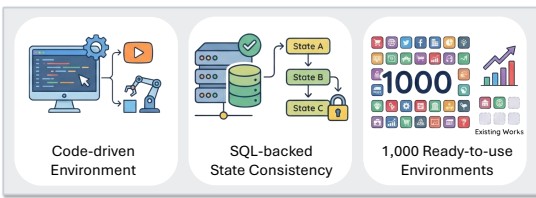

*Figure 1.* Agent World Model (AWM) is a synthetic environment generation pipeline that synthesizes 1,000 diverse code-driven agentic environments with databases for training tool-use agents.

OpenAI, 2025; Anthropic, 2025a; Comanici et al., 2025; Guo et al., 2025). Agents powered by LLMs emerge as a promising paradigm for handling multi-step complex tasks in realistic environments (Nakano et al., 2021; Yao et al., 2023; Yang et al., 2024; Qin et al., 2024). However, training such agents often requires performing large-scale reinforcement learning (RL) on diverse environments that are relatively resource-scarce and expensive to scale.

Using real-world environments for training is prohibitively expensive and hard to scale, since many scenarios do not expose public APIs and RL training often requires agents to interact with them thousands of times in a stable and efficient manner (Dulac-Arnold et al., 2021; Qin et al., 2024; Xu et al., 2024a; Luo et al., 2025b). Human-created environments are hard to scale and often lack diversity. For example, $\tau^2$-bench (Barres et al., 2025) and TheMCPCompany (Esfandiarpoor et al., 2025) only contain three and five environments, respectively, which is far from enough for training generic AI agents. Most existing synthetic research on agents focuses on task synthesis (Wang et al., 2023; Chen et al., 2024; Xie et al., 2025; Patil et al., 2024; Wang et al., 2026) and trajectory collection (Xu et al., 2024b; Li et al., 2025a; Song et al., 2024) rather than environment synthesis. Another line of research simulates the tool response or even the environment, where each state transition is generated by LLMs (Liu et al., 2024b; Lu et al., 2025; Li et al., 2025b;c; Chen et al., 2025). However, this approach is not reliable or efficient due to the hallucination issue (Wang et al., 2024; Kalai et al., 2025) and LLM's high inference cost.

These limitations highlight a missing piece: scalable environment synthesis. In particular, The challenge is to synthesize executable, reliable environments at scale, enabling

*These authors contributed equally and share last authorship. Work done during Zhaoyang Wang's internship at Snowflake. [1]University of North Carolina at Chapel Hill [2]Snowflake. Correspondence to: Canwen Xu <cxu@ucsd.edu>.

*Proceedings of the 43rd International Conference on Machine Learning*, Seoul, South Korea. PMLR 306, 2026. Copyright 2026 by the author(s).

replicable agent interaction and learning. In recent months, DeepSeek-V3.2 (DeepSeek-AI et al., 2025) introduces a synthesis pipeline to create thousands of executable environments for general agents, while Qwen Tongyi (Fang et al., 2025) also describes an environment synthesis pipeline but for supervised fine-tuning (SFT) rather than RL training. The adopted code-based approach is promising to build such environments at scale, since it can control the state transition and ensure the consistency of the environment. However, neither of them releases the generation pipeline nor open-sources their environments. Several concurrent community efforts (Feng et al., 2025; Sullivan et al., 2025; Cai et al., 2025; Zhang et al., 2025a; Song et al., 2026) explore environment synthesis through programming. However, they either target game-like environments, rely on human priors (e.g., code documentation), lack strong guarantees of state consistency, or remain limited in scale.

To address this, we propose Agent World Model (**AWM**), an open-source pipeline that synthesizes executable tool-use environments at scale. AWM is analogous to learned world models in model-based RL but realized via code-driven environments rather than neural dynamics. By decomposing synthesis into these three components, we can leverage LLMs to generate each part systematically while maintaining consistency. Our AWM mirrors how software is built in practice. Starting from a high-level scenario description (e.g., "an online shopping platform"), we first generate common user requirements (i.e., tasks) that users are likely to perform in this scenario. Then, we generate the database schema to define what entities and relations exist to fulfill these user requirements. This schema can guide the design of exposed interfaces (toolset) and help generate the backend code for the interfaces, ensuring each tool has a clear data model to operate on. The interface is exposed via Model Context Protocol (MCP) (Anthropic, 2024) for unified agent tool interaction with the environment. Finally, we generate verification code that compares the database state before and after agent execution, which augments an LLM-as-a-Judge to provide robust reward signals for RL training. Critically, each stage includes automated execution and simple self-correction: if generated code fails to run, we feed the error information back to the LLM for correction.

This pipeline enables us to scale to 1,000 unique environments spanning most real-world scenarios such as shopping, social media, finance, and travel. Each environment provides an executable sandbox where agents can interact with dozens of tools, and they fully support parallel isolated instances and are easy to reset or restart, which are important for efficient online RL. To validate AWM, we perform large-scale RL (each step with 1,024 environment instances) to train agents to use MCP tools to complete the task. In contrast to recent work that trains agents using benchmark-specific environments or simulators conditioned on benchmark task sets (Li et al., 2025b; Chen et al., 2025; Luo et al., 2025a; Prabhakar et al., 2025; Wang et al., 2025b), our environments and tasks are not tailored for any benchmarks or specific scenarios. Experimental results on three tool-use benchmarks demonstrate the generalization performance of our framework. In summary, our contributions are three-fold: (1) AWM, an open-source pipeline for automatic generation of executable tool-use environments with database-backed state consistency, (2) 1,000 ready-to-use diverse environments suitable for large-scale RL training, and (3) empirical results show that agents trained on AWM generalize to out-of-distribution unseen environments.

## 2. Related Work

**Tool-use Agents.** Recent work shows that LLMs can use external tools to solve complex tasks (Qin et al., 2024; OpenAI, 2025; DeepSeek-AI et al., 2025; Team et al., 2025). Toolformer (Schick et al., 2023) trains tool-use LLMs by supervised learning. ToolLLM (Qin et al., 2024) curates real-world APIs and trains on LLM-generated trajectories, but uses simulated responses instead of tool execution. Gorilla (Patil et al., 2024) fine-tunes with API documentation to improve tool-use accuracy. ReAct (Yao et al., 2023) and SWE-agent (Yang et al., 2024) alternate reasoning and acting in interactive environments. However, most training data remains static or comes from small-scale environments. Existing benchmarks (Yao et al., 2024; Barres et al., 2025; Esfandiarpoor et al., 2025; Luo et al., 2025b; Fan et al., 2025; Mo et al., 2025) either use real-world APIs or provide small-scale environments. This makes them hard to use as large-scale RL training grounds. The gap is the lack of diverse and executable environments that are efficient enough for RL which needs extensive agent interactions, and it benefits from fast interactions and reliable state transitions.

**Agent Data Synthesis.** Synthetic data is widely used to scale agent training (Patil et al., 2024; Schick et al., 2023; Qin et al., 2024; Liu et al., 2024b; 2025). Self-Instruct (Wang et al., 2023) popularizes using LLMs to generate data for fine-tuning. Subsequent work uses LLMs to synthesize tasks, tools specifications, and agent trajectories (Xie et al., 2025; Xu et al., 2024b; Li et al., 2025a; Song et al., 2024; Liu et al., 2024b; 2025; Wang et al., 2026; Prabhakar et al., 2025). These methods provide diverse tasks and tool-use demonstrations, but do not offer executable environments for agent interaction. The key limitation is that they treat the environment as given or simulate tool responses with LLMs. Without environment synthesis, agents cannot explore alternative actions or receive grounded feedback from state changes, which limits applicability to RL.

**Environment Synthesis.** As agentic RL grows, the need for diverse and executable environments becomes more urgent. Environment synthesis largely follows two directions:

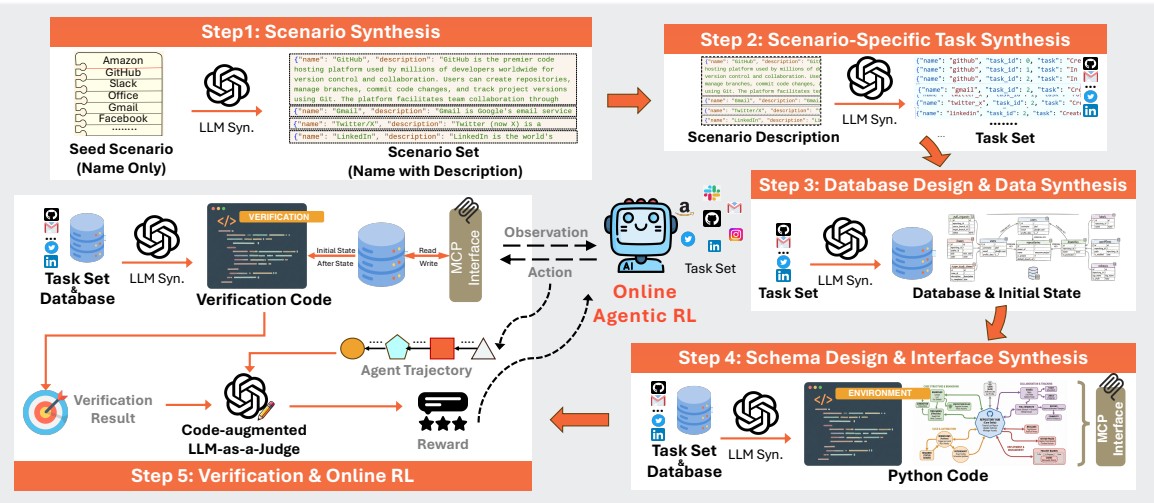

*Figure 2.* Overview of AWM. Starting from scenario synthesis, we progressively generate tasks, database, interface and verification to obtain fully executable environments. Then, we perform multi-turn RL training for tool-use agents in our synthesized environments.

LLM-based simulation (Wang et al., 2024; Li et al., 2025c; Chen et al., 2025; Li et al., 2025b) and programming-based synthesis (Tang et al., 2024; Sullivan et al., 2025; Cai et al., 2025; Zhang et al., 2025a; Song et al., 2026). LLM-based simulation often simulates the environment by prompting reasoning model to generate state transition and observation. However, it suffers from hallucinations in state transition (Kalai et al., 2025; Wang et al., 2024). It is also expensive and inefficient for RL, since each environment step may require an LLM call. Programming-based synthesis builds environments through programming and database, where each state transition and observation are driven by the code. Some methods extract tool graphs from documentation and sample tool call sequences to build environments (Fang et al., 2025; Cai et al., 2025). Procedural generation has also been explored, for example with human designed type systems (Sullivan et al., 2025). AutoEnv (Zhang et al., 2025a) creates 36 game-like environments (e.g., maze navigation) for simulating heterogeneous worlds. EnvScaler (Song et al., 2026) is a concurrent work that synthesizes 191 interactive environments via code generation given existing task sets.

AWM differs in three aspects: (1) We synthesize from scratch without assuming predefined task sets or API documentation, which avoids human priors. (2) We use database-backed state management to enforce consistency and enable code-augmented verification for RL. (3) With 1,000 environments, 35,062 tools and 10,000 tasks paired with verification code synthesized, to the best of our knowledge, AWM is the largest open-source environment set to date.

## 3. Agent World Model

In this section, we describe AWM in details, as shown in Figure 2. We synthesize environments as partially observable

Markov decision processes (POMDPs) suitable for agentic RL training. Each environment $E_i$ comprises five components: a state space $\mathcal{S}_{E_i}$, an action space $\mathcal{A}_{E_i}$, an observation space $\mathcal{O}_{E_i}$, a transition function $T_{E_i} : \mathcal{S}_{E_i} \times \mathcal{A}_{E_i} \to \mathcal{S}_{E_i} \times \mathcal{O}_{E_i}$, and task-specific reward functions $R_\tau$ for each $\tau \in \mathcal{T}_{E_i}$ that is the task set for the environment. The goal of the agent is to complete tasks by interacting with the environment through multi-turn tool interactions. The synthesis pipeline in Sec. 3.3.1 would instantiate: the **database** defines $\mathcal{S}_{E_i}$, the **interface** layer defines $\mathcal{A}_{E_i}$, $\mathcal{O}_{E_i}$, and $T_{E_i}$, while **verification** provides $R_\tau$ for each user task.

### 3.1. Scenario Generation

We leverage the vast world knowledge of modern LLMs to generate diverse scenario descriptions (websites, apps, or common tools collection), seeded with 100 popular domain names. Unlike general web agents that navigate static content sites (e.g., news, wikis), we focus on stateful applications (e.g., e-commerce, CRM, management) that necessitate database interactions instead of information retrieval. To ensure quality, we employ a filtering pipeline: an LLM-based classifier selects scenarios involving core CRUD operations (create, read, update, delete), while embedding-based deduplication ensures diversity. We also cap over-represented categories to prevent the collection from collapsing to a few dominant types. This process yields 1,000 unique scenarios spanning finance, travel, retail, social media, workflow, user management, and more, as shown in Figure 6 and Table 15 in Appendix A.1.

### 3.2. Task Generation

Following software engineering principles, we then synthesize user tasks to serve as functional requirements for the en-

vironment. These tasks would dictate the necessary database entities and API endpoints in subsequent synthesis steps. For each scenario, we prompt the LLM to generate $k = 10$ different tasks $\mathcal{T}_{E_i} = \{\tau_{i,j}\}_{j=1}^{k}$ covering diverse functionalities of the scenario. We enforce two design principles: (1) *API-solvability*, avoiding purely UI-dependent actions (e.g., clicking, page navigation), and (2) *post-authentication context*, assuming login is completed to focus on deep functionalities rather than access control, because such authentication is typically handled by human in realistic settings. This results in 10,000 executable tasks that would drive the synthesis of the environment backends.

### 3.3. Environment Synthesis

Given a scenario description and its task set $\mathcal{T}_{E_i}$, we synthesize an executable environment $E_i$ by instantiating POMDP components. First, we construct the state space $\mathcal{S}_{E_i}$ and initial state by generating a SQLite (Gaffney et al., 2022) schema and populating it with synthetic sample data that supports the entities and constraints implied by $\mathcal{T}_{E_i}$. Then, we generate an MCP (Anthropic, 2024) interface layer that exposes a toolset in Python, which defines the action space $\mathcal{A}_{E_i}$ as tool calls and the observation space $\mathcal{O}_{E_i}$ as tool responses. Calling a tool runs database operations that read and write, which implements the environment transition function $T_{E_i}$. Finally, we synthesize verification logic to define the task-specific reward functions $R_\tau$. This design grounds environment state in structures, while ensuring agents interact with $E_i$ only via a unified MCP interface.

#### 3.3.1. ENVIRONMENT

**Database.** The database grounds each environment in a concrete and persistent state. We use SQLite (Gaffney et al., 2022) as the state backend for structured relational state, unlike the simplified NoSQL or key-value stores used in concurrent works (Zhang et al., 2025a; Cai et al., 2025; Song et al., 2026). Given the scenario and task set $\mathcal{T}_{E_i}$, the LLM infers required entities/attributes/relations to make every task feasible, generating tables only when required by the tasks. We exclude authentication-related fields to align with the task generation step. The schema defines the state space $\mathcal{S}_{E_i}$ and constrains all transitions through explicit keys and constraints. However, schema alone is insufficient, as many tasks require querying or updating existing records. We therefore synthesize sample data that instantiates a realistic initial state $s_0$, ensuring every task in $\mathcal{T}_{E_i}$ is executable from the start. The LLM analyzes preconditions for each task and creates records satisfying these constraints, including variation data needed for robust execution.

**Interface.** Agents cannot directly manipulate the database without breaking abstraction. We introduce a Python interface layer exposed via MCP (Anthropic, 2024) that defines the action space $\mathcal{A}_{E_i}$ and observation space $\mathcal{O}_{E_i}$. We use two stages: first toolset schema design, then code generation. Pilot experiments show that environments may require over 3,000 lines of code, making direct generation challenging without schema guidance. Given the task set and database schema, the LLM infers the minimal set of operations required to make every task executable, generating endpoints only when necessary. The schema also serves as documentation for agents through summaries, typed parameters, and response schemas, as shown in Table 16. We then generate an executable Python file where each endpoint becomes an MCP tool. Tool execution triggers database operations, implementing the transition function $T_{E_i}$.

**Verification.** To complete the POMDP specification, we define task-specific reward functions $R_\tau$ to enable RL training. We design a task-associated verification module for each task $\tau$ that grounds evaluation in the environment state. Specifically, this module inspects the database state before and after agent execution, extracting task-relevant signals and success or failure criteria that describe how to interpret state differences. However, because our environments are fully synthetic, verification can occasionally be affected by environment imperfections, such as incomplete state updates, unexpected execution failures, or infrastructure-related issues (e.g., timeouts). To improve reward robustness, the ultimate decision is made by an LLM-as-a-Judge (Zheng et al., 2023), which combines the agent trajectory with structured verification signals. LLM-as-a-Judge complements code-based checks by leveraging trajectory-level context which helps mitigate misjudgments caused by imperfect environment signals. Finally, the verification step returns one of {`Completed`, `Partially Completed`, `Agent Error`, `Environment Error`}.

A natural question arises: *why not rely entirely on code-driven verification*? While appealing, this approach assumes that task success is perfectly specifiable and reliably observable from state alone. In practice, this assumption is fragile. Even realistic services exhibit imperfect behavior due to transient failures, partial executions, or infrastructure issues; synthetic environments are no exception. In general task settings, brittle verification logic can introduce false positives or negatives, an issue that has surfaced repeatedly in existing benchmarks such as WebArena, $\tau^2$-bench, and SWE-Bench, which were later revised to correct earlier misjudgments (hattami et al., 2025; Cuadron et al., 2025; OpenAI, 2024). Our code-augmented LLM-as-a-Judge addresses this by combining the precision of code-based verification with the flexibility and context-awareness of LLM reasoning. Specifically, structured verification signals ground the LLM in concrete evidence, enabling it to resolve ambiguities that rigid code alone cannot handle. We also prepare the code-driven verification for AWM, providing the community with both options, though our analysis in Sec. 6.2 and case study

*Table 1.* Statistics of the synthesis pipeline. Success rates and trial counts reflect the self-correction. Costs are averaged per 100 samples using GPT-5 (OpenAI, 2025) as the generation model.

| Synthesis Stage | Success (%) | # Trial | Cost ($) |
|---|---|---|---|
| Scenario | – | – | 0.43 |
| Task | – | – | 0.56 |
| Database | 88.3 | 1.12 | 3.59 |
| Sample Data | 88.2 | 1.12 | 13.75 |
| Toolset Schema | – | – | 23.74 |
| Env Code | 86.8 | 1.13 | 12.81 |
| Verification | – | – | 2.21 |
| **Total** | – | – | **57.09** |

*Table 2.* Environment complexity statistics. Agent steps and unique tools are measured using Claude-4.5-Sonnet (Anthropic, 2025a) as the agent backbone, with a step budget of 20. The completion rate is $62.6\%$, and $13.7\%$ of tasks exceeded the step budget.

| Metric (#) | Mean | Median | Top 90% |
|---|---|---|---|
| Database Tables | 18.5 | 18.0 | 25.0 |
| Sample Data Records | 129.3 | 121.0 | 192.0 |
| Exposed Tools | 35.1 | 35.0 | 45.0 |
| Environment Code Lines | 1,984.7 | 1,944.0 | 2,586.0 |
| Agent Steps per Task | 8.5 | 6.0 | 20.0 |
| Unique Tools used per Task | 7.1 | 6.0 | 12.0 |

*Table 3.* Comparison with existing programming-based environments. AWM generates environments with minimal reliance (only a seed set of scenario names) on human or real APIs, achieving the largest scale with SQL-backed state consistency. A direct empirical comparison with AutoForge is not feasible because its code and environments are not released.

| Method | Syn. | Reliance | SQL | # Envs | # Tools | # Code |
|---|---|---|---|---|---|---|
| $\tau$-bench | × | Human | × | 2 | 12.5 | – |
| $\tau^2$-bench | × | Human | × | 3 | 22.7 | – |
| MCP-Universe | × | Real APIs | – | 11 | 12.1 | – |
| AutoForge | ✓ | Tool Doc | × | 10 | – | – |
| EnvScaler | ✓ | Task Set | × | 191 | 18.6 | 662.1 |
| **AWM** | ✓ | Names Only | ✓ | **1,000** | **35.1** | **1984.7** |

of verification failure cases in Appendix B.4 confirm that code-augmented LLM-as-a-Judge is more robust.

**Execution-based Self-Correction.** Across all synthesis steps described above, we employ a simple self-correction mechanism to handle generation errors. After generating each component of the environment, we attempt to run it in an isolated environment and test the functionality. If any error occurs (e.g., runtime exceptions), we capture the error message and feed it back to the LLM together with the problematic code snippet, prompting the LLM to regenerate a corrected version. This process is repeated for up to five iterations or until the component executes successfully. In practice, we often find that this lightweight retry strategy is effective in repairing generated code, without requiring a more complex correction mechanism. During the synthesis process, we finally allow up to $10\%$ errors in each stage to reduce costs, while acknowledging that setting a stricter threshold or using modern coding agent harness such as Claude Code can further improve the quality of the environments at the cost of more correction iterations.

### 3.3.2. PIPELINE RESULTS AND ANALYSIS

Through AWM, we synthesize 1,000 executable environments along with 10,000 tasks. Table 1 shows the statistics of the synthesis process, and Table 2 reports their complexity, indicating that each stage produces non-trivial artifacts far beyond toy environments. The pipeline achieves over $85\%$ first-attempt success rate, with the self-correction mechanism requiring only 1.13 iterations on average to repair generation failures, suggesting that most failures are shallow runtime issues that can be fixed with small edits. The statistics further show that each stage generates non-trivial artifacts, far exceeding toy environments. Table 3 compares AWM with existing environment sets: our method achieves the largest scale, with $5\times$ more environments than the closest concurrent work EnvScaler (Song et al., 2026), while requiring minimal human participation beyond 100 scenario names. This shows that AWM is feasible for large-scale cost-effective synthesis of executable environments.

## 4. Agentic Reinforcement Learning

With the synthesized environments, we perform online reinforcement learning for tool-use agents using Group Relative Policy Optimization (GRPO) (Shao et al., 2024). Agentic interaction involves long-horizon trajectories with interleaved observations and tool calls, requiring both careful reward design and alignment between training and inference.

### 4.1. Reward Design

Purely outcome-based rewards have shown success in mathematical reasoning (Guo et al., 2025); however, in agentic environments, they may be insufficient and inefficient to regularize tool-use behavior. We therefore adopt a hybrid reward design that combines step-level format correctness with task-level outcome verification. At each step $t$, we check whether the tool call follows the required format (see Appendix A.4). Any violation triggers early termination of the rollout and an immediate negative reward. This both discourages invalid actions and saves computation in long-horizon multi-turn settings. After a rollout terminates normally, we evaluate the task-level outcome via our code-augmented LLM-as-a-Judge, defining the final reward as:

$$R_\tau = \begin{cases} 1.0, & \text{if task } \tau \text{ Completed,} \\ 0.1, & \text{if task } \tau \text{ Partially Completed,} \\ 0.0, & \text{otherwise.} \end{cases} \quad (1)$$

The step-level reward $r_t$ follows: if early termination occurs at step $t$, $r_t = -1.0$; if the rollout terminates normally,

$r_t = R_\tau$ is broadcast to all action steps; otherwise $r_t = 0$. This design encourages syntactically valid tool usage while preserving outcome-driven optimization.

## 4.2. History-Aware Training

When deploying agents, history context is often managed by a dedicated framework that strategically truncates long interaction histories to avoid attention sink and improve efficiency (Liu et al., 2024a; Anthropic, 2025b; Zhang et al., 2025b). However, existing RL training pipelines may optimize policies using full histories, creating a distribution mismatch issue between training and inference.

Let $h_t = (o_1, a_1, o_2, a_2, \ldots, o_t)$ denote the full interaction history. In practice, many RL frameworks (Sheng et al., 2024; Hu et al., 2025) optimize all actions from a completed rollout in one model forward pass for efficiency:

$$
\mathcal{L} = -r \log \pi_\theta (\underbrace{o_1, a_1, o_2, a_2, \ldots, o_T, a_T}_{\text{one forward pass}})
$$
$$
\cdot \underbrace{[0, 1, 0, 1, \ldots, 0, 1]}_{\text{loss mask}}, \tag{2}
$$

where $\pi_\theta$ is the agent parameterized by $\theta$, and the mask selects action tokens while ignoring observations. At inference time, however, the agent may condition on a truncated history $h_t^{\text{trunc}} = (o_{\max(1, t-w+1)}, a_{\max(1, t-w+1)}, \ldots, o_t)$, which results in a distribution shift if training always uses full history. To address this issue, we align training with inference by applying the same truncation during optimization. Under GRPO, for each task $\tau$ in environment $E_i$, we sample a group of $G$ rollout trajectories $\{y^{(k)}\}_{k=1}^G$, where $y^{(k)} = (a_1^{(k)}, \ldots, a_{T_k}^{(k)})$, and optimize:

$$
\mathcal{L}_{\text{GRPO}} = \mathbb{E}_{\tau, E_i, \{y^{(k)}\}} \left[ \frac{1}{G} \sum_{k=1}^G A^{(k)} \sum_{t=1}^{T_k} \log \pi_\theta(a_t^{(k)} \mid h_t^{\text{trunc},(k)}) \right], \tag{3}
$$

where $A^{(k)} = (R^{(k)} - \bar{R})/\sigma_R$ is the group-relative advantage computed from the rollout rewards $\{R^{(j)}\}_{j=1}^G$. This objective splits the trajectory into multiple individual sub-trajectories, each conditioned on its own truncated history, ensuring consistency with inference-time execution.

## 5. Experiments

### 5.1. Experimental Setup

**Benchmarks.** We evaluate agents on three tool-use and MCP benchmarks to assess out-of-distribution generalization: (1) The verified version of $\tau^2$-bench (Barres et al., 2025; Cuadron et al., 2025), which consists of multi-turn conversational agentic tasks across three representative scenarios: airline, retail, and telecom; (2) BFCLv3 (Patil et al., 2025), a comprehensive benchmark for function-calling

ability evaluation, covering single-turn, multi-turn (long-context), synthetic tools, real-world tools, and hallucination tests, resulting in four evaluation categories: non-live, live, multi-turn, and hallucination; (3) MCP-Universe (Luo et al., 2025b), a collection of real-world MCP servers spanning location navigation, financial analysis, browser automation, web search, and multi-server workflows. We exclude 3D design tasks that require GUI and repository management tasks that require authenticated access to GitHub or Notion.

**Baselines.** We compare against the following baselines: (1) Base: the original LLMs equipped with reasoning and tool-use capabilities, without additional training; (2) Simulator (Li et al., 2025b; Chen et al., 2025): agents trained with RL in LLM-simulated environments, where GPT-5 (OpenAI, 2025) serves as the environment transition model, using the same tasks and toolsets as AWM, to highlight the advantage of executable environments over simulated ones; (3) EnvScaler (Song et al., 2026): a concurrent method that synthesizes 191 programming-based environments for SFT and RL, included to compare the synthesis quality.

**Implementation Details.** The AWM pipeline is instantiated using GPT-5 (OpenAI, 2025), which is also used as the code-augmented LLM-as-a-Judge for task verification. We select Qwen3 thinking models (Yang et al., 2025) across 4B, 8B, and 14B scales as base agents due to their popular community adoption. Multi-turn RL training is implemented on top of AgentFly (Wang et al., 2025a) and verl (Sheng et al., 2024). Due to limited computation budget, we train agents on a subset of AWM consisting of 526 environments and 3,315 tasks. Each model is trained for up to 96 optimization steps with a fixed learning rate of $7 \times 10^{-7}$. We use a batch size of 64 and 16 rollouts, resulting in 1,024 isolated environment instances launched per step. The sliding window size and maximum interaction turn are set to $w = 3$ and 20, respectively. For evaluation, we adapt benchmarks to MCP interface by requiring agents to use tools via two unified tools: `list_tools` and `call_tool`. More implementation details can be found in Appendix A.

### 5.2. Main Results

Table 4 presents results on three out-of-distribution benchmarks. AWM does not target conversational interaction, whereas $\tau^2$-bench requires multi-turn dialogue. AWM omits refusal scenarios, while BFCLv3 stresses hallucination resistance. AWM also excludes browser automation and information retrieval, both central to MCP-Universe.

On BFCLv3, AWM improves performance across all models. For 8B model, the overall score increases from 53.83 to 65.94, surpassing Simulator and EnvScaler. Gains are broadly distributed, with a modest weakness on hallucination, likely due to our format correctness reward that always encourages tool use and penalizes refusals (Appendix A.4).

*Table 4.* Results across BFCLv3, $\tau^2$-bench, and MCP-Universe benchmarks. Best performance is in **bold**. For BFCLv3, "Hall." refers to the hallucination category. For $\tau^2$-bench, Pass@k denotes task success rate allowing $k$ attempts, with Pass@1 averaged over 4 runs. For MCP-Universe, we report task success rates. Note that EnvScaler did not release the 14B model.

| | BFCLv3 Leaderboard | | | | |
|---|---|---|---|---|---|
| Method | Non-Live | Live | Multi-Turn | Hall. | Overall |
| **4B** | | | | | |
| Base | 61.44 | 63.95 | **39.38** | 73.93 | 54.92 |
| Simulator | 66.10 | 62.69 | 37.75 | 75.48 | 55.52 |
| EnvScaler | 60.31 | 64.25 | 37.63 | **85.25** | 54.06 |
| AWM | **78.60** | **76.91** | 37.99 | 73.70 | **64.50** |
| **8B** | | | | | |
| Base | 59.58 | 58.03 | 43.88 | 76.42 | 53.83 |
| Simulator | 56.35 | 59.36 | 41.88 | 74.06 | 52.53 |
| EnvScaler | 33.67 | 31.83 | **45.00** | **90.30** | 36.83 |
| AWM | **80.44** | **72.39** | **45.00** | 70.80 | **65.94** |
| **14B** | | | | | |
| Base | 69.48 | 67.28 | 47.00 | 77.94 | 61.25 |
| Simulator | 77.23 | 73.43 | **52.38** | 76.91 | 67.68 |
| AWM | **81.46** | **77.20** | 51.88 | **78.37** | **70.18** |

| | $\tau^2$-Bench | | | | |
|---|---|---|---|---|---|
| | Domain (Pass@1) | | | Overall | |
| Method | Airline | Retail | Telecom | Pass@1 | Pass@4 |
| **4B** | | | | | |
| Base | 21.00 | 19.96 | 9.43 | 15.83 | 34.89 |
| Simulator | 15.50 | 18.64 | 7.90 | 13.67 | 35.25 |
| EnvScaler | **31.50** | **44.30** | 12.50 | **28.96** | **51.80** |
| AWM | 19.00 | 30.26 | 16.45 | 22.57 | 43.89 |
| **8B** | | | | | |
| Base | 26.50 | 34.43 | 18.42 | 26.44 | 50.72 |
| Simulator | 34.00 | 32.24 | 29.17 | 31.30 | 54.32 |
| EnvScaler | 31.50 | **49.56** | **32.68** | **39.39** | **63.31** |
| AWM | **38.50** | 41.23 | 23.47 | 33.45 | 55.40 |
| **14B** | | | | | |
| Base | 27.00 | **65.35** | 12.28 | 36.69 | 55.40 |
| Simulator | 21.50 | 48.90 | **18.20** | 31.39 | 55.40 |
| AWM | **31.50** | 63.60 | 17.76 | **39.03** | **57.19** |

| | MCP-Universe | | | | | |
|---|---|---|---|---|---|---|
| Method | Location | Financial | Browser | Web | Multi | Overall |
| **4B** | | | | | | |
| Base | **2.86** | **25.00** | 0.00 | 0.00 | 0.00 | 6.15 |
| Simulator | 0.00 | 15.00 | **5.88** | 2.00 | 0.00 | 6.15 |
| EnvScaler | 0.00 | 17.50 | 2.94 | 0.00 | 0.00 | 4.47 |
| AWM | 0.00 | 22.50 | 2.94 | 2.00 | 5.00 | **6.70** |
| **8B** | | | | | | |
| Base | 0.00 | 22.50 | 5.88 | 2.00 | 0.00 | 6.70 |
| Simulator | 5.71 | 10.00 | **8.82** | 4.00 | 0.00 | 6.15 |
| EnvScaler | 0.00 | 17.50 | 5.88 | 2.00 | 0.00 | 5.59 |
| AWM | 8.57 | 35.00 | **8.82** | 0.00 | 5.00 | **11.17** |
| **14B** | | | | | | |
| Base | 0.00 | 27.50 | 5.88 | 2.00 | **5.00** | 8.38 |
| Simulator | 2.86 | 30.00 | 8.82 | **6.00** | 0.00 | 10.62 |
| AWM | 8.57 | 32.50 | **11.77** | 4.00 | 0.00 | **12.29** |

*Table 5.* Benchmark task complexity stratification with 8B agent. MCP-Universe does not have ground truth trajectory reference or other complexity indicators, thus being excluded.

| | BFCLv3 (by # tool calls) | | | $\tau^2$ (by # GT actions) | | |
|---|---|---|---|---|---|---|
| Bucket | Base | AWM | $\Delta$ | Base | AWM | $\Delta$ |
| Simple | 53.6 | **80.3** | +26.7 | 32.7 | **41.9** | +9.3 |
| Medium | 60.0 | **75.3** | +15.3 | 22.7 | **28.8** | +6.2 |
| Hard | 43.9 | **45.0** | +1.1 | 20.5 | **25.0** | +4.5 |

*Table 6.* Quality analysis of 100 sampled environments. We compare AWM with EnvScaler using two LLM judges (GPT-5.1 and Claude-4.5-Sonnet). Top section shows scores (1–5 scale, ↑ higher is better). The bottom shows code bug analysis (↓ lower is better). Note that bugs may only affect a few tools and edge use cases.

| | GPT-5.1 | | Claude 4.5 | |
|---|---|---|---|---|
| Metric | AWM | EnvScaler | AWM | EnvScaler |
| *LLM-as-a-Judge Scores (↑)* | | | | |
| Task Feasibility | **3.68**±1.02 | 2.94±1.25 | **3.99**±0.81 | 3.14±1.29 |
| Data Alignment | **4.04**±0.91 | 3.73±0.89 | **4.84**±0.50 | 4.11±1.02 |
| Toolset Completeness | **3.65**±0.87 | 2.89±0.79 | **4.98**±0.14 | 4.06±0.87 |
| *Bug Analysis (↓)* | | | | |
| Envs w/ Bugs | **74%** | 88% | 83% | 83% |
| # Bugs per Env | 4.13 | **1.82** | 2.70 | **2.21** |
| Blocked Tasks | **14.0%** | 57.1% | **11.5%** | 46.8% |

On $\tau^2$-bench, AWM is competitive with EnvScaler and consistently exceeds Simulator. Notably, EnvScaler regresses on BFCLv3 (-8.93) and MCP-Universe (-1.39) on average, whereas ours improves over Base across all benchmarks; plausibly because EnvScaler relies on existing tasks for synthesis that may overlap with $\tau^2$-bench. On MCP-Universe, AWM achieves the best overall results, with large gains in Financial and Location. These results indicate that training on our synthetic environments builds robust tool-use capabilities that transfer to real-world scenarios. Moreover, the comparison with Simulator suggests that programming-based state consistency provides a more stable learning

signal than LLM-generated interactions, while substantially reducing RL latency, since Simulator requires an LLM call at each interaction step. Overall, AWM shows the strongest generalization, validating the effectiveness of our synthesis.

**Gains across complexity.** We stratify BFCLv3 and $\tau^2$-bench tasks by inherent difficulty (number of required tool calls / ground-truth actions) in Table 5. AWM improves over Base across all complexity buckets on both benchmarks. The absolute gain shrinks on harder tasks while the relative gain remains substantial, which indicates the agent's performance on hard tasks relies on its intrinsic capabilities.

## 6. Analysis

### 6.1. Quality of Synthesized Environments

We evaluate synthesized environments along three axes that are crucial for agentic RL training: *quality* (tasks are executable with coherent environment), *difficulty* (synthesized tasks bring new information), and *diversity* (training does not collapse to near-duplicate environments).

**Quality.** Table 6 reports LLM-as-a-Judge scores on Task Feasibility, Data Alignment, and Toolset Completeness, where AWM consistently outperforms EnvScaler under both GPT-5.1 and Claude-4.5-Sonnet judges, indicating stronger end-to-end consistency from tasks → database → interface. Despite our environments containing roughly 3x more code than those generated by EnvScaler (Table 3), this increase

*Table 7.* Quality analysis of AWM synthesis pipeline on 100 environments (1,000 tasks) under three different generator models.

| Metric \ Generator | GPT-5 | Claude-4.5 | Qwen3.5 |
|---|---|---|---|
| *Pipeline Success Rate (%, ↑)* | | | |
| Database Synthesis | 88.3 | 100.0 | 79.0 |
| Sample Data Synthesis | 88.2 | 100.0 | 97.0 |
| Env. Code Synthesis | 86.8 | 99.0 | 77.0 |
| *Quality Score (1–5, ↑)* | | | |
| Task Feasibility | 3.99 | 3.84 | 3.15 |
| Data Alignment | 4.84 | 4.92 | 4.32 |
| Toolset Completeness | 4.98 | 4.97 | 4.30 |
| *Bug Statistics (↓)* | | | |
| Envs w/ Bugs (%) | 83 | 89 | 100 |
| # Bugs per Env | 2.70 | 2.11 | 3.82 |
| *Diversity (↑)* | | | |
| Mean Pairwise Distance | 0.34 | 0.31 | 0.35 |

*Table 8.* Complexity-stratified Pass@1 (%) evaluation on synthesized tasks by GPT-5.1 and Claude-4.5-Sonnet. Buckets are defined by required tool calls: 1-3, 4-6, 7-10 and 11+.

| Metric \ Bucket | Simple | Medium | Hard | Very Hard |
|---|---|---|---|---|
| Tasks Ratio (%) | 35.1 | 31.5 | 15.7 | 17.7 |
| Avg. Steps | 3.3 | 5.8 | 9.1 | 17.9 |
| GPT-5.1 | 61.7 | 27.0 | 11.9 | 3.0 |
| Sonnet-4.5 | 68.4 | 71.8 | 63.3 | 31.0 |
| Both Pass | 45.3 | 24.9 | 11.5 | 3.0 |
| Neither Pass | 15.2 | 26.2 | 36.3 | 69.0 |

in scale results in only a moderate rise in bugs, highlighting effective scalability. Both methods exhibit implementation issues at this scale: manual inspection on our environments attributes 44% of bugs to unhandled edge input cases (e.g., missing null/boundary validation) and 14% to operations conflicting with database constraints (e.g., foreign-key or uniqueness violations), which together account for the majority of bugs and are typical of auto-generated code at this scale. During RL training, the environment error rate consistently remains low, around 4%, most of which are internal errors caused by specific agent behaviors, indicating overall reliability and usability of our environments. AWM yields fewer blocked tasks than EnvScaler, which is critical for RL because blocked tasks truncate exploration and inject systematically incorrect negative signals. To mitigate such imperfect-environment issues, we design the `Environment Error` reward in Sec. 3.3.1.

To verify the universality of our pipeline, we further test Claude-4.5-Sonnet and open-source Qwen3.5-122B-A10B for the synthesis in Table 7. The pipeline is largely model-agnostic: Claude-4.5 attains 99% environment-code success and matches GPT-5 on quality, while Qwen3.5 still reaches

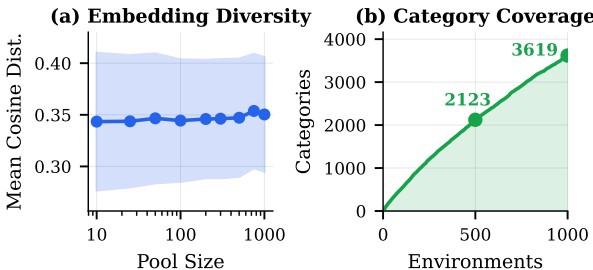

*Figure 3.* Diversity analysis of 1,000 synthesized environments. (a) Embedding diversity is calculated by encoding the scenario description, database schema and toolset schema. (b) Category coverage counts the number of unique topics of scenarios.

77% with usable but bug-heavier environments. Diversity is essentially constant across generators, confirming that it comes from the pipeline design rather than the used model.

**Difficulty.** We present a complexity-stratified analysis of AWM tasks in Table 8. Overall, the two advanced models achieve only 36.1% and 62.6% Pass@1 respectively, confirming that AWM tasks are non-trivial (GPT-5.1's lower rate may be partly due to its occasional refusal to follow the system prompt). Pass rates decrease monotonically with complexity for both models, and 69% of Very-Hard tasks are unsolved by either. Furthermore, from our RL training logs, 27.1%, 23.7%, and 28.2% of tasks are never solved even once by Qwen3-4B/8B/14B throughout the entire training process (14B only optimizes for 32 steps), providing further evidence that AWM generates genuinely challenging tasks that remain difficult even after extensive RL exploration.

**Diversity.** Figure 3 analyzes diversity from semantic and topical perspectives. Embedding diversity remains stable as pool size grows, suggesting newly generated environments continue to add novel content rather than forming duplicates. Meanwhile, category coverage steadily increases, showing that AWM globally expands into new regions instead of collapsing to a few dominant domains. We further conduct a code-level similarity study using AST-based duplicate detection and token/n-gram Jaccard metrics. The cross-environment AST function duplicate rate is 0.0% and endpoint name Jaccard is 0.004 (Appendix B.3), confirming that diversity holds even at the implementation level.

### 6.2. Analysis on Verification Design

Table 9 compares three verification strategies. LLM-only verification is ungrounded in state changes and yields the weakest performance. Code-only verification improves over it but is brittle under environment imperfections, which can produce false negatives. Our code-augmented strategy combines structured state-diff signals with a reasoning LLM, achieving the best results across all model scales and

*Table 9.* Overall performance across three verification strategies: LLM-only, code-only, and code-augmented. LLM-only verification decides the rewards based on the agent trajectory. Code-only verification inspects database state differences and the final answer, deciding either `Completed` ($r_t = 1$) or `Others` ($r_t = 0$).

| Size | Verification | BFCLv3 | $\tau^2$ P@1 | $\tau^2$ P@4 | MCP |
|------|--------------|--------|--------|--------|------|
| 4B | LLM | 51.92 | 15.65 | 35.97 | **6.70** |
| | Code | 55.66 | 14.93 | 32.01 | 6.15 |
| | Augmented | **64.50** | **22.57** | **43.89** | **6.70** |
| 8B | LLM | 55.46 | 26.44 | 52.52 | 10.62 |
| | Code | 60.00 | 29.59 | 52.88 | 5.59 |
| | Augmented | **65.94** | **33.45** | **55.40** | **11.17** |
| 14B | LLM | 65.44 | 31.03 | 55.76 | 10.62 |
| | Code | 65.04 | 29.50 | 53.60 | 8.38 |
| | Augmented | **70.18** | **39.03** | **57.19** | **12.29** |

*Table 10.* Code-augmented LLM-as-a-Judge reliability on 100 sampled trajectories evaluated 5 times each. Here we report the binary classification results (`Completed` vs. others).

| Metric | GPT-5.1 | Sonnet-4.5 | Qwen3.5 |
|--------|---------|------------|---------|
| Self-Consistency ($\uparrow$) | 90.8% | 82.0% | 76.3% |
| Pairwise Agreement ($\uparrow$) | 95.5% | 91.8% | 88.1% |
| Fleiss' $\kappa$ ($\uparrow$) | 0.891 | 0.826 | 0.728 |
| Reward Flip Rate ($\downarrow$) | 9.2% | 18.0% | 23.7% |

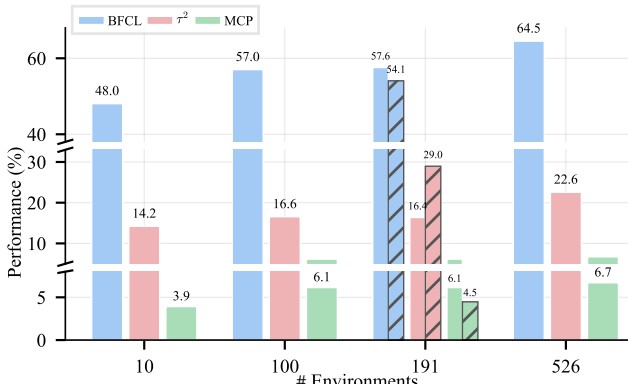

*Figure 4.* Scaling curve of AWM over environment quantity with the 4B model. Hatched bars denote EnvScaler which synthesized 191 environments for SFT and RL training.

*Table 11.* Analysis of history-aware training with aligned and misaligned inference settings with 4B model. "Aligned" uses the same history limit (HL) setting for training and inference (i.e., no truncation for model w/o HL, while truncated history for model w/ HL). "Misaligned" uses opposite settings for training and inference.

| Setting | Method | BFCLv3 | $\tau^2$ P@1 | $\tau^2$ P@4 | MCP |
|---------|--------|--------|--------|--------|------|
| Aligned | w/ HL | 64.50 | 22.57 | 43.89 | 6.70 |
| | w/o HL | 55.35 | 15.92 | 36.33 | 6.15 |
| Misaligned | w/ HL | 61.85 | 9.35 | 15.11 | 5.03 |
| | w/o HL | 56.80 | 16.10 | 33.09 | 6.15 |

benchmarks. The extra cost of GPT-5 as the judge is about $1.80 per training step (at most 1,024 samples), and the asynchronous setting introduces negligible latency. We also study the format correctness reward in Appendix B.1.

**Judge reliability.** To verify the judge reliability for RL, we sample 100 trajectories and judge each 5 times with three different LLMs in Table 10. GPT-5.1 attains 95.5% pairwise agreement and Fleiss' $\kappa$ of 0.891 with only a 9.2% reward flip rate, justifying the high reliability for RL training. Even open-source Qwen3.5-122B-A10B model reaches 88.1% agreement, and the high cross-model agreement indicates that our designed verification rubrics and signals capture genuine task completion rather than model-specific ones. More details are provided in Appendix B.2.

### 6.3. Analysis on History-Aware Training

Table 11 compares our history-aware training against a full-history variant (w/o HL) under aligned and misaligned inference. When aligned, AWM w/ HL achieves the best results among all settings, indicating the benefit of optimizing directly under truncated histories. In contrast, the full-history variant is relatively insensitive to misalignment. When truncated at inference, it even shows a slight improvement in $\tau^2$, consistent with truncation suppressing interference from irrelevant turns. These findings suggest that agent history context management should be part of policy optimization rather than a purely inference-time heuristic.

### 6.4. Environment Scaling Curve

Figure 4 studies the scaling curve with respect to the number of training environments. Training on 10 environments leads to severe performance degradation, likely due to overfitting to the limited distribution. Scaling to 100 environments yields substantial gains, and at a matched count, AWM outperforms EnvScaler on both BFCLv3 and MCP-Universe benchmarks. Further scaling to 526 environments continues to improve agent performance. This monotonic improvement highlights the scalability of AWM for agentic RL.

## 7. Conclusion

In this paper, we propose Agent World Model (AWM), a scalable pipeline that synthesizes executable agentic environments by mirroring software development. Using AWM, we successfully scale to 1,000 environments with 10,000 tasks. These environments are code-driven and backed by SQL databases exposed through a unified MCP interface, supporting parallel isolated instances for large-scale agentic RL. Experiments on three benchmarks show that agents trained on our synthetic environments generalize well to out-of-distribution domains, outperforming both LLM-simulated training and concurrent synthesis methods. We believe AWM's scalable pipeline and its 1,000 synthesized environments are valuable resources to the community.

## Impact Statement

This work presents an open-source pipeline for synthesizing executable environments to train tool-use agents. By releasing both the generation pipeline and the synthesized environments, we aim to lower the barrier for the research community to study agentic systems. However, synthetic environments may not fully reflect real-world scenarios, thus we encourage the community to apply safeguards when deploying agents trained on AWM to real-world scenarios.

## Limitations

**Opportunities for Self-Evolving.** Our AWM pipeline synthesizes environments through a fixed generation process, inherently limiting the model's ability to autonomously improve and evolve beyond the initial capabilities. While this approach has shown improvements in our experiments, incorporating a self-evolving paradigm where a trained agent contributes to synthesizing new environments allows the model to continuously adapt and improve its capabilities. We leave this as a promising direction for future work.

**Synthesis Pipeline Optimization.** Current AWM pipeline provides opportunities for further optimization. For example, self-correction currently operates primarily through trial-and-error, addressing runtime errors without deeper semantic validation. Integrating an LLM to proactively detect logical inconsistencies or subtle bugs not caught by runtime checks could enhance the robustness and semantic coherence of generated environments. It is also valuable to take human inspection to further improve the quality of synthesized environments if resources are available. Moreover, synthesizing tasks that span multiple scenarios or environments represents another promising area for extension. Both enhancements are readily compatible with our existing architecture, offering clear pathways for future improvements.

**Training Scale and Model Coverage.** Due to limited computing resources, we train agents on 526 of the 1,000 synthesized environments. We also mainly focus on Qwen3 model family (Yang et al., 2025) across 4B, 8B, and 14B scales. Despite these limitations, the experiments already demonstrate consistent out-of-distribution generalization across three benchmarks, suggesting that even a subset of AWM provides a strong training signal. Importantly, the core contribution of this paper is the synthesis pipeline itself. The released environments and pipeline are model-agnostic and can benefit the broader community regardless of the specific models used in our experiments.

**Robustness & Safety.** AWM only exposes agents to noise arising from imperfect synthesis (74% of environments contain at least one bug, ∼4% of RL rollouts hit runtime errors), and does not evaluate behavior under intentionally injected adversarial perturbations such as corrupted database rows, malicious tool outputs, or prompt-injection content in observations. Since task generation assumes a post-authentication context for tractability, access-control behavior is also not directly trained, thus an over-eager agent could in principle perform unauthorized data access, irreversible state mutations, or unintended call chains. Some synthesized scenarios further touch sensitive domains (e.g., finance and healthcare) where errors carry asymmetric costs, despite AWM containing no real personal data. We therefore view adversarial robustness evaluation, access control training, and domain specific human-in-the-loop oversight as prerequisites for production deployment.

## Use of AI Assistants

We acknowledge the use of AI assistants in the writing of this paper, primarily for polishing text and creating figure icons. All generated content is reviewed by the authors, and necessary corrections are made.

## Acknowledgment

We would like to thank Jeff Rasley for helping organize AWM's related open-source resources.

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

# A. Implementation Details

## A.1. Scenario Generation

We adopt a Self-Instruct style expansion (Wang et al., 2023) to scale from 100 seed websites to 1,000 diverse scenarios. The seed set is drawn from popular domain names[1], and the LLM generates new candidates conditioned on few-shot examples that emphasize stateful, database-backed interactions. To maintain diversity, we apply two complementary filters: (1) an LLM classifier that scores each candidate on suitability for CRUD operations (create, read, update, delete), rejecting content-centric or read-only scenarios (e.g., news), and (2) embedding-based deduplication using cosine similarity with a threshold of 0.85, ensuring newly generated scenarios are sufficiently distinct from the existing pool. We also enforce category caps to prevent over-representation of dominant types such as e-commerce. The prompts used for generation and classification are provided in Figures 10 and 11. Table 15 shows a random sample of 100 generated scenarios. We also show the distribution of the synthesized scenarios in Figure 6 and the wordcloud of the scenario descriptions in Figure 7. The distribution shows that the synthesized scenarios span diverse domains without collapsing to a few dominant types.

## A.2. Task Generation

For each scenario, we prompt the LLM to generate $k = 10$ diverse user tasks that serve as functional requirements for downstream synthesis. This modest count keeps the database schema and toolset implementation tractable for LLMs. The prompt enforces two constraints: (1) tasks must be API-solvable without UI dependencies such as clicking or page navigation, and (2) tasks assume post-authentication context to unlock deeper functionalities. Each task is required to be specific and self-contained, including concrete parameters (e.g., product IDs, user names) so that it can be executed without additional clarification. This requirement-driven design ensures that the subsequently generated database schema and toolset are aligned with actual user needs rather than being over-specified or incomplete. The prompt template is shown in Figure 12. And the generated tasks are shown in Table 17.

## A.3. Environment Synthesis

This section details the synthesis of the three core POMDP components: state space (database), action and observation spaces (interface), and reward function (verification).

**Database Schema Generation.** Given the scenario description and task set $\mathcal{T}_{E_i}$, the LLM infers the minimal relational schema required to make all tasks feasible. The output is a set of SQLite DDL statements defining tables, columns, types, primary keys, foreign keys, and indexes. We instruct the LLM to reason about entity relationships implied by the tasks (e.g., a "cancel order" task implies an Orders table with a status column) and to avoid authentication-related fields. Based on the introduced self-correction mechanism, we attempt to run the generated DDL statements in an isolated environment and test the functionality. If any DDL statement fails execution, we capture the error message, summarize it via a separate LLM call, and append the summary to the prompt for regeneration. This feedback loop repeats with an error threshold of 10%: if fewer than 10% of tables fail, we accept the schema. The prompt is shown in Figure 13. And we visualize part of the SQLite database schema with sample data in Figure 8.

**Sample Data Synthesis.** An empty schema is insufficient for task execution, since many tasks require querying or updating existing records. We synthesize sample data by prompting the LLM to analyze task preconditions and generate INSERT statements that satisfy them. For example, if a task requires "updating product inventory," the synthesized data must include at least one product with a non-zero stock level. The LLM outputs a JSON structure containing table names and corresponding INSERT statements, which are executed in dependency order respecting foreign key constraints. We apply the same error-feedback loop as schema generation, with an error threshold of 10% for acceptable insert failures. The prompt is shown in Figures 14 and 15.

**Interface Specification Design.** Before generating implementation code, we first synthesize an interface specification that defines the toolset schema. This two-stage approach (schema then code) reduces hallucination in long code generation, since pilot experiments show that direct code generation for environments with 30+ tools often produces inconsistent interfaces. Given the task set and database schema, the LLM infers the minimal set of endpoints required to make every task executable. Each endpoint specification includes: name, method, path, summary, typed input parameters, and response schema. The specification also serves as documentation for agents at inference time, shown in Table 16. The prompt is

---

[1] https://www.similarweb.com/top-websites/

shown in Figures 16 and 17, and an example of the synthesized MCP toolset is shown in Figure 25.

**Environment Implementation.** With the API specification and database schema, we generate a complete Python file implementing an MCP server. The generated code includes: SQLAlchemy ORM models mirroring the database schema, Pydantic request/response models, and FastAPI endpoint handlers that perform database operations. Each environment averages about 2,000 lines of code and exposes 35 tools on average. After generation, we validate the environment by launching and checking that the MCP server starts successfully and responds to health checks and "list tools" calls. Failed environments enter the self-correction loop to fix the errors. The prompt is shown in Figures 18, 19, and 20.

**Verification Code Synthesis.** For each task, we generate a Python verification function that compares database states before and after agent execution. The function takes two database paths (initial and final) as input, executes SQL queries to extract task-relevant information, and returns a structured dictionary containing: changed records, expected outcomes, and diagnostic signals. We also generate success and failure criteria in natural language that guide the LLM-as-a-Judge in interpreting the verification results. During RL training, this code-augmented verification provides grounded evidence to the judge, reducing hallucination in reward assignment. The prompt for generating the verification code is shown in Table 21, and the example of the synthesized verification code is shown in Figure 26 and 27.

**Self-Correction Mechanism.** All synthesis stages share a common error-recovery pattern. When generated code fails execution, we capture the full error traceback and invoke a separate LLM call to produce a concise error summary (typically 200-500 tokens) that identifies the root cause and suggests fixes. This summary is appended to the original prompt for the next generation attempt. We limit retries to 5 iterations per component. If an error threshold (10% for schema/data, 0% for environment startup) is exceeded after all retries, we select the best attempt based on error rate and proceed. This approach achieves over 85% first-attempt success rate across all stages, with an average of 1.13 correction iterations for failed cases.

## A.4. Tool Calling Format and Validation

We adopt the Qwen3 tool-calling format (Yang et al., 2025), which wraps each tool invocation in XML tags: "`<tool_call>`{name: ..., arguments: ...}`</tool_call>`". Rather than exposing all environment-specific tools directly to the agent, we design a two-level abstraction that decouples the agent from environment details.

**Two-Level Tool Abstraction** The agent interacts with MCP environments through exactly two meta-tools: (1) `list_tools`, which queries the MCP server to retrieve all available tools in the current environment along with their metadata (input/output schemas, descriptions), and (2) `call_tool`, which invokes an environment-specific tool by name with the required arguments passed as a JSON string. This design enables the agent to dynamically discover and invoke different toolsets across environments without hardcoding any tool information. The complete system prompt is shown in Figure 9.

**Format Validation Rules** During RL training, we enforce format constraints to ensure well-formed tool calls and prevent hallucinated or malformed interactions. We apply rule-based validation at each step of the trajectory, classifying it as either valid, a format error, or an environment/server error. The validation checks the following conditions: (1) Reasoning format: All assistant messages must contain non-empty reasoning within `<think>...</think>` tags. (2) Tool name validity: The agent must not call hallucinated tools (tools not returned by `list_tools`) or use invalid tool names. (3) Argument validity: Tool arguments must be well-formed JSON that conforms to the tool schema. (4) Protocol adherence: The agent must call `list_tools` exactly once, and it must be the first tool call in the trajectory. (5) Interaction consistency: If the agent produces multiple interaction turns, it must make at least one successful tool call beyond the initial `list_tools`. (6) Server response: Each tool call must receive a non-error response from the MCP server. For example, if the server returns an error message indicating timeout or internal failure, we classify it as an environment error. For each step $t$, if any of the 1-5 conditions are violated, we classify it as a format error, assigning a negative reward $r_t = -1$. If condition 6 is violated, we classify it as an environment error with reward $r_t = 0$ following the reward shapes in Sec. 4.1.

**Early Termination** To improve training efficiency, we implement early stopping for unrecoverable errors. When the validator detects a format error or server error during rollout, we terminate the trajectory immediately rather than continuing to generate additional steps. This saves computational resources and prevents the agent from receiving reward signals for fundamentally broken trajectories.

## A.5. Training and Evaluation Details

**Training Hyperparameters.** Table 12 summarizes the hyperparameters used for GRPO training (Shao et al., 2024). According to the common agentic RL training settings (Li et al., 2025b; Wang et al., 2025b), we set the KL coefficient to

0.001. Beyond standard GRPO settings, we use a higher clip ratio of 0.28, following the recommendation in DAPO (Yu et al., 2025) to allow more exploration for the agent. Also, we use sequence-level importance sampling to mitigate the distribution shift issue between the rollout engine and the model training engine (Yao et al., 2025).

**Environment Management.** We implement multi-turn RL training on top of AgentFly (Wang et al., 2025a) and verl (Sheng et al., 2024). Each training step launches 1,024 isolated environment instances in parallel, with each instance running as an independent MCP server backed by its own SQLite database copy. This isolation ensures that concurrent rollouts do not interfere with each other's state transitions. Environment instances are reset by restoring the database to its initial state after each rollout completes. Environment startup (spawning MCP servers, copying databases) is a bottleneck in online RL, as it blocks rollout collection. To overlap environment preparation with policy training, we further implement a pre-fetching mechanism: while the current batch undergoes gradient updates, a background thread pre-configures environments for the next batch. This significantly reduces per-step wall-clock time compared to sequential environment startup.

**Sample Splitting for History-Aware Training.** As described in Section 4.2, we align training with inference by applying history truncation during optimization. We use simple history context management, specifically sliding window truncation, to study the mismatch issue between training and inference. Specifically, given a completed rollout with $T$ assistant turns, we split it into $T$ separate training samples. For sample $t$, the input consists of the system prompt, initial user message, the first assistant-tool exchange (which contains the `list_tools` call), and the $w = 3$ most recent turns preceding turn $t$. The loss is computed only on the tokens of turn $t$, while all preceding context tokens have their loss mask set to zero. This "sample splitting" approach ensures each training sample mirrors the truncated context the agent will see at inference time. Although this increases the number of forward passes per rollout by a factor of $T$, it eliminates the distribution shift that would otherwise occur when training on full histories but deploying with truncated contexts. More complex history context management is possible, but it is beyond the scope of this paper.

**Reward Computation.** At the end of each rollout, we invoke the code-augmented LLM-as-a-Judge to determine task completion status. The verification code is executed on the final database state, comparing it against the initial state, and its structured output (changed records, success criteria) is provided to GPT-5 (OpenAI, 2025) along with the agent trajectory. The judge returns one of four classifications: `Completed`, `Partially Completed`, `Agent Error`, or `Environment Error`. We map these to rewards as specified in Sec. 4.1. To reduce latency, verification calls are batched and executed asynchronously while the next rollout batch is being collected, adding negligible overhead to the training loop.

**Evaluation Protocol.** We evaluate trained agents on three benchmarks that differ substantially from our training distribution. A key challenge is that each benchmark uses a different tool-calling format: $\tau^2$-bench (Barres et al., 2025; Cuadron et al., 2025) expects direct tool names (e.g., `get_user_details`), BFCLv3 (Patil et al., 2025) uses function-calling syntax, and MCP-Universe (Luo et al., 2025b) uses the MCP protocol. To bridge this gap, we implement format converters that translate between our unified `call_tool` abstraction and each benchmark's native format. For $\tau^2$-bench, we wrap each tool call as `call_tool(tool_name="mcp_tool_{name}", arguments={...})` and unwrap responses accordingly. For BFCLv3, we aggregate multi-turn trajectories into the expected single-turn or parallel function call format, skipping `list_tools` meta-calls. This ensures evaluation measures the agent's actual task-solving ability rather than format compliance.

**Decoding and Context Configuration.** During evaluation, we use temperature 0.6 with top-$k$=20 and top-$p$=0.95, following the recommended decoding settings in Qwen3 (Yang et al., 2025). We extend the context window to 131,072 tokens via RoPE scaling (Su et al., 2024) for all evaluations, as some benchmark tasks (especially BFCLv3 multi-turn categories) require processing lengthy interaction histories. When processing long context tasks, the history limit is loosened to $w = 10$ turns during evaluation, allowing agents to leverage more context information.

# B. Analysis

## B.1. Analysis on Step-level Format Correctness Reward

We study the impact of the step-level format correctness reward in Figure 5. With this reward, agents quickly learn to follow the tool interface contract, and the format error ratio rapidly converges to a low level. It also improves training efficiency by reducing the average rollout time by about 27%. In contrast, without this reward, the format error ratio remains above 20% even after 50 optimization steps, leading to frequent invalid actions and noisier learning signals. This degradation further translates into a lower task completion rate, which saturates below 40%. Overall, the step-level format correctness reward improves both agent performance and training efficiency.

*Table 12.* Hyperparameters for GRPO training.

| Hyperparameter | Value |
|---|---|
| *GRPO* | |
| Learning rate | $7 \times 10^{-7}$ |
| Batch size | 64 |
| Mini-batch size | 16 |
| Rollouts per task $G$ | 16 |
| Instances per step | 1,024 |
| Max optimization steps | 96 |
| KL coefficient | 0.001 |
| Entropy coefficient | 0.0 |
| Clip ratio (high) | 0.28 |
| *Rollout* | |
| Temperature | 1.0 |
| Max response length | 2,048 |
| Max model context | 32,000 |
| *Agent* | |
| Max interaction turn | 20 |
| History window size | 3 |

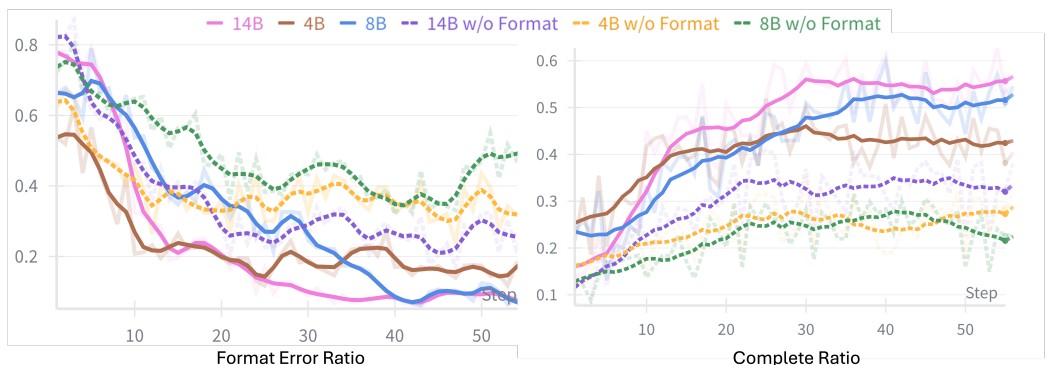

*Figure 5.* Format error ratio comparison of AWM. "w/o Format" means disabling the step-level format correctness reward.

## B.2. Detailed LLM-Judge Reliability

We elaborate on the reliability study summarized in Sec. 6.2. The full results are shown in Table 13. We sample 100 agent trajectories from the 8B agent during training and judge each trajectory 5 times with each of three judges (GPT-5.1, Claude-4.5-Sonnet, Qwen3.5-122B-A10B), using default or recommended temperature of each model. We compute four metrics: *Self-Consistency* (fraction of trajectories on which a single judge returns the same label in all 5 trials), *Pairwise Agreement* (fraction of pairs of trials that agree), *Fleiss' $\kappa$*, and *Reward Flip Rate* (fraction of trajectories where at least one trial disagrees with the majority). Even under the strictest 4-class setting, GPT-5.1 attains 91.2% pairwise agreement, and the open-source Qwen3.5 reaches 82.7%. This high reliability validates the use of our code-augmented LLM-as-a-Judge for reward assignment during RL training.

## B.3. Code-Level Diversity Analysis

To complement the embedding-based diversity analysis in Figure 3, we measure cross-environment similarity directly at the code level for all 1,000 environments in Table 14. We use the AST-based Tree-Structure Edit Distance (TSED) implementation[2] to detect duplicated function bodies across files, and token-level / endpoint-name / class-name Jaccard for lexical comparison. The results show that no function-level structural duplicates exist across any pair of environments at TSED threshold 0.5, endpoint and class names have cross-environment Jaccard well below 0.01, and token Jaccard stays

---

[2] https://github.com/mizchi/similarity

*Table 13.* Detailed LLM-judge reliability on 100 sampled trajectories evaluated 5 times each by three different judges. We report two settings: binary classification (`Completed` vs. others) which is the actual RL reward signal, and detailed 4-class classification across all reward labels (`Completed`/`Partially Completed`/`Agent Error`/`Environment Error`).

| Metric | GPT-5.1 | Claude-4.5-Sonnet | Qwen3.5-122B-A10B |
|---|---|---|---|
| *Binary (`Completed` vs. others)* | | | |
| Self-Consistency (↑) | 90.8% | 82.0% | 76.3% |
| Pairwise Agreement (↑) | 95.5% | 91.8% | 88.1% |
| Fleiss' $\kappa$ (↑) | 0.891 | 0.826 | 0.728 |
| Reward Flip Rate (↓) | 9.2% | 18.0% | 23.7% |
| *4-class (full reward classification)* | | | |
| Self-Consistency (↑) | 82.7% | 65.0% | 69.5% |
| Pairwise Agreement (↑) | 91.2% | 83.5% | 82.7% |
| Fleiss' $\kappa$ (↑) | 0.781 | 0.693 | 0.650 |
| Reward Flip Rate (↓) | 11.2% | 25.0% | 18.6% |

*Table 14.* Code-level similarity across AWM environments. We measure structural similarity via AST-based Tree-Structure Edit Distance (TSED), and lexical similarity via token / endpoint / class-name Jaccard.

| Metric | Granularity | Mean Similarity | Max |
|---|---|---|---|
| AST Function Duplicate (TSED, threshold $\geq$ 0.5) | function-level | 0.0% | 0.0% |
| Token Jaccard | token-level | 0.183 | 0.324 |
| Endpoint Name Jaccard | API route-level | 0.004 | — |
| Class Name Jaccard | schema-level | 0.009 | — |

low (max 0.324 and the residual may come from shared Python/FastAPI keywords). The code-level evidence corroborates our embedding-level diversity finding: AWM pipeline is suitable for large-scale agent environments synthesis.

### B.4. Case Study for Verification

We provide three representative cases in Figures 28, 29, and 30 to illustrate why our code-augmented LLM-as-a-Judge is more robust than either rigid code-only verification or LLM-only judging from agent trajectories. For the first case (Fig. 28), the task is a clean, database-grounded query: the agent calls the correct tool to retrieve the full bid history of the item and summarize the findings; the verifier can deterministically confirm these statistics from the underlying state/returned records, and the judge aligns, illustrating the ideal regime where structured verification evidence is decisive. For the second case (Fig. 29), the environment exhibits an imperfection when the agent tries to create a routine; a strict code-only verifier that expects a state delta would incorrectly flag failure because the initial and final snapshots appear identical, yet the trajectory shows an idempotent success path, and the code-augmented judge uses this context to correctly mark `Completed`, reducing false negatives under transient tool/infrastructure issues. For the last case (Fig. 30), an API/tool calling error misleads the agent into creating a duplicate event and then adding the session under the wrong event ID; tool calls succeed locally so a judge without verifier grounding could be fooled into marking `Completed`, but the verifier reveals the real target event remains unchanged, enabling the code-augmented judge to identify such wrong operation.

Overall, the key theme is that synthetic interactive environments are imperfect: transient tool failures, idempotent tasks, and ambiguous tool calling behaviors can all break the assumptions of rigid verification. Our design mitigates this by letting the verifier extract *structured evidence* from database snapshots and rule-based checks, while the judge reasons over *trajectory context* to resolve ambiguity and reduce both false negatives and false positives.

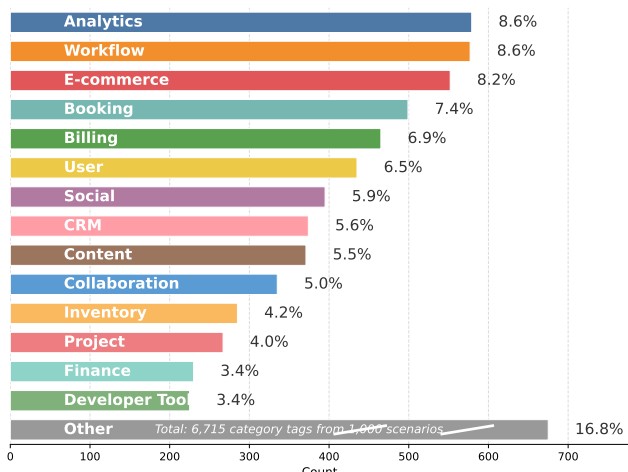

Figure 6. Distribution of synthesized scenarios of AWM.

Figure 7. Wordcloud of the scenario descriptions of AWM.

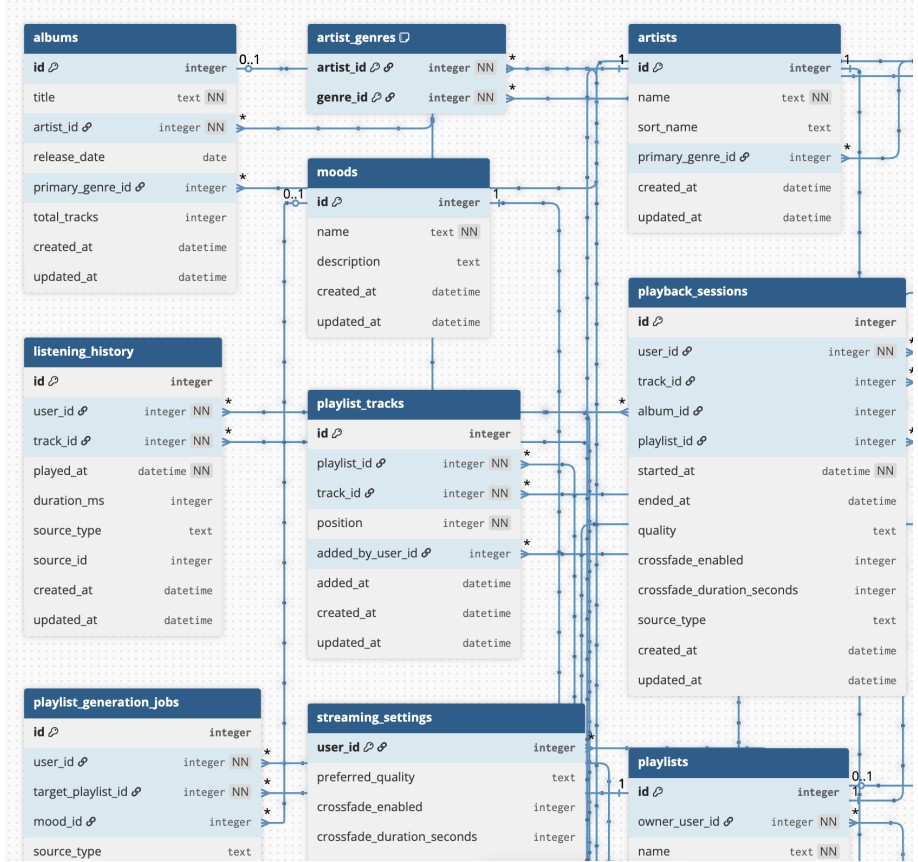

Figure 8. Visualization of part of the SQLite database schema of "Spotify" environment.

*Table 15.* A random sample of 100 generated scenarios from AWM. The collection includes both synthetic applications and real-world services that represent common enterprise and consumer tool-use patterns.

| Scenario Name | Scenario Name | Scenario Name |
| --- | --- | --- |
| ArenaPlay Competitive Gaming Hub | AT&T | AutoCareLane Service Scheduler |
| AutoGrid Dealer Inventory Manager | Basecamp | Bill.com |
| BlockBridge HOA & Condo Community Manager | CampusEnroll Admissions Registration Suite | Canva |
| Charles Schwab | CineRealm Digital Cinema | ClaimWise Property & Auto Insurance Claims Desk |
| ClassBench Music School Scheduling & Attendance | ClassTrack Micro-LMS | ClinicConnect Multi-Specialty Appointment Center |
| ClinicPaws Veterinary Practice Suite | ClubNest Membership Hub | CodeMentor Classroom LMS |
| CraftBench Custom Manufacturing Job Tracker | DeskQueue IT Support | DevForum Hub (Technical Q&A Forum) |
| DeviceFleet Cloud Appliance Manager | Discord | Duo Security (Cisco Duo) |
| Eventbrite | EventHarbor Conference & Session Manager | FleetAxis Commercial Fleet Command |
| Fleetio | FleetPath Manager | FlexDesk Workspace Reservations |
| FlowMesh Workflow Automation Hub | FlowPay Subscription Billing Engine | FlowTrack Team Workflow Hub |
| FormPilot Enterprise Surveys & Workflows | FundHarbor Retail Investing & Portfolio Tracker | GhostKitchenOS Virtual Brand Manager |
| GitHub | GiveStream Community Grants Hub | GiveWell-style Donation Portal |
| Gmail | GrantPath Scholarship Application Portal | GreenLedger Utility & Sustainability Dashboard |
| GuildForge Esports Hub | HelpLane Support Desk | HireBench Coding Graduate Pipeline |
| HireBoard Remote Jobs Marketplace | HomeFlow Smart Device Console | HomeGrid Smart Panel |
| HostEase Property Bookings | InsightLoop Feedback & NPS Tracker | Jira |
| KitchenGrid Restaurant Ops Hub | LeaseLink Corporate Workspace | ListNest Shared Collections & Bookmarks |
| LootPlaza Virtual Goods Marketplace | MacroMate Corporate Meal Program Portal | Microsoft 365 (Office) |
| Microsoft Account (Live.com) | Mindbody | NeighborHands Volunteer Network |
| NeighborLink Mutual Aid Exchange | Notion | OpenTable |
| PantryPicks Grocery & Recipe Wishlist | PawSitter Network & Booking Hub | PayBridge Merchant Payments Gateway |
| PayStream Merchant Payment Hub | PetCrate Supply Subscriptions | PinHarbor Visual Bookmarks |
| PrepLine Kitchen Display & Routing | QuickBite Campus Food Ordering | QuickLift Auto Service Booking |
| RegShield Policy Compliance & Attestation Center | Restaurant Backoffice Pro | Reverb |
| Salesforce | ShelfStack Personal Media Library | SignSure Enterprise E-Signature & Policy Acknowledgment |
| SitBuddy Pet Sitting & Home Visit Marketplace | SkinMart Virtual Goods Marketplace | SlotBand Event & Vendor Scheduler |
| StackList Personal Collections Manager | StockCrate Warehouse Inventory Cloud | Stripe Dashboard |
| SubScript Subscription & Billing Manager | SubStream Creator Membership Hub | SyncRoom Remote Collaboration Suite |
| TailCart Pet Supplies Marketplace | TalentLoop Recruiting CRM | TeleVisit360 Virtual Care |
| Toast POS | TutorLane Subject Sessions | VendorVault Procurement & Vendor Portal |
| VenueLane Event Space Booker | VetConnect Telehealth Hub | VolunteerGrid Scheduling & Shift Manager |
| VolunteerShift Scheduler | WalkLoop Pet Sitting Network | WishCart Social Shopping Lists |
| WishLoom Multi-Store Gift Registry | | |

*Table 16.* Example of interface schema snippet with rich annotations and explicit database constraints.

```
"operation_id": "get_product_by_id",
"request_params": {"product_id": {"type": "integer", "required": true}},
"response": {
  "product": {"id": {"type": "integer"}, "title": {"type": "string"}, ....},
  "aggregates": {"average_rating": {"type": "float"}, ....}, ....},
"required_tables": ["products", "product_aggregates", "product_offers"],
"required_fields": {
  "products": ["id", "title", "category_id", ....], ....}
```

```
# MCP Tools

You are at a MCP environment.  You need to call MCP tools to assist with the
user query.  At each step, you can only call one function.  You have already
logged in, and your user id is 1 if required for the MCP tool.

You are provided with TWO functions within <tools></tools> XML tags:
<tools>
1.  list_tools
- Description:  List all available MCP tools for the current environment to
help you finish the user task.
- Arguments:  None
- Output:  A list of MCP environment-specific tools and their descriptions

2.  call_tool
- Description:  Call a MCP environment-specific tool
- Arguments:
- tool_name:  str, required, the tool name in the list_tools output
- arguments:  str, required, the arguments for calling <tool_name>.  You must
pass a valid JSON string without any markdown fences or additional commentary.
This JSON str will be parsed by the tool and executed.  You can pass an empty
JSON str if no arguments are required by <tool_name>.
- Output:  The result of the <tool_name> tool call
</tools>

You should always call list_tools function first to get the available tools,
and should only call it once.  You should always directly output the answer or
summary at the final step instead of calling any function.

For each function call, return a json object with function name and arguments
within <tool_call></tool_call> XML tags:
<tool_call>
{"name":  <function-name>, "arguments":  <args-json-object>}
</tool_call>

Example Function Call #1:
<tool_call>
{"name":  "list_tools", "arguments":  null}
</tool_call>

Example Function Call #2:
<tool_call>
{"name":  "call_tool", "arguments":  {"tool_name":  "get_weather", "arguments":
"{"city":  "Beijing"}"}}
</tool_call>
```

*Figure 9.* System prompt for agent in MCP environments.

```
You are an expert at identifying websites, apps, and digital platforms that are HIGHLY
SUITABLE for API environment simulation and database-driven interactions.

## KEY PRINCIPLE: Can the data be SYNTHESIZED?
We need platforms where data can be FAKED but still be REALISTIC and USEFUL.

### Data that CAN be synthesized:
- Numbers:  temperatures, prices, quantities, ratings, coordinates, stock prices
- Entities:  users, products, orders, bookings, employees, devices, sensors
- Status values:  order status, flight status, device state
- Short text:  names, titles, addresses, categories, descriptions
- Timestamps:  dates, schedules, deadlines, historical records
- Geographic:  cities, airports, coordinates (static data)

### Data that CANNOT be synthesized meaningfully:
- Long articles:  news content, blog posts, encyclopedia entries
- Media:  actual video/audio content (not metadata)
- AI inference:  chatbot responses, ML-based recommendations
- Real search:  search engine results with ranking

## HIGH Suitability Examples (GENERATE THESE):
E-commerce:  CRUD on orders, reviews (products, orders, cart, reviews)
Task Management:  CRUD on tasks (tasks, boards, assignments)
Banking/Fintech:  CRUD on transactions (accounts, transactions, transfers)
Booking/Reservation:  CRUD on reservations (listings, bookings, availability)
Weather Services:  Query weather data (locations, forecasts, alerts, history)
Flight/Travel:  Search & book flights (flights, airports, bookings, prices)
Stock Trading:  Trade & portfolio (stocks, trades, portfolio, prices)
IoT/Smart Home:  Control devices (devices, sensors, readings, schedules)
Fitness Tracking:  Log workouts (workouts, exercises, metrics, goals)
Healthcare:  Manage appointments (patients, appointments, records, prescriptions)
HR/Payroll:  Manage employees (employees, timesheets, payroll)
CRM/Sales:  Manage leads (leads, deals, contacts, activities)
Inventory:  Manage stock (products, inventory, warehouses)
Logistics:  Track shipments (shipments, packages, routes, status)
Restaurant/Food:  Orders & menu (menu_items, orders, reservations)
Property/Real Estate:  Listings & viewings (properties, listings, viewings, offers)
Education/LMS: Courses & grades (courses, enrollments, assignments, grades)
Event Management:  Events & tickets (events, tickets, attendees, venues)

## LOW Suitability (AVOID THESE):
News sites (BBC, CNN): Article CONTENT is the product (cannot synthesize meaningful
articles)
Wikipedia/Encyclopedia:  Article CONTENT is the product (cannot fake knowledge)
Search Engines:  Need real search index + ranking (cannot simulate relevance)
AI Assistants (ChatGPT): Need real AI inference (cannot fake AI responses)
Translation (Google Translate):  Need real NLP models (cannot fake translations)
Content platforms focus:  If CONTENT is core value (metadata is not enough)

## MEDIUM Suitability (Focus on CRUD aspects):
YouTube:  CAN simulate playlists, subs, comments, channel mgmt; CANNOT simulate actual
videos, recommendations
Spotify:  CAN simulate playlists, library, following; CANNOT simulate actual audio,
music discovery
Reddit:  CAN simulate posts (short), comments, votes; CANNOT simulate long-form quality
content

## Categories to Cover (for diversity):
E-commerce, Booking/Reservation, Task/Project Management, Finance/Banking,
Weather/Environmental, Flight/Travel, Stock/Investment,
IoT/Smart Home, Fitness/Health Tracking, Healthcare/Medical, HR/Recruiting, CRM/Sales,
Inventory/Warehouse, Logistics/Shipping,
Restaurant/Food, Real Estate, Education/LMS, Event/Ticketing, Legal/Contracts,
Subscription Management, Customer Support,
Forms/Surveys, Analytics Dashboards, Fleet Management, Utility Management (electricity,
water), Insurance, Pet Services, Automotive

## Your Task:
Generate NEW platforms that are HIGHLY SUITABLE where data can be realistically
SYNTHESIZED and users perform meaningful CRUD operations.
```

*Figure 10.* System prompt for scenario generation (part 1 of 2).

```
Here are {num_examples} examples of suitable website/app scenarios:

{examples}

---

Now generate {num_to_generate} NEW and DIVERSE website/app/platform scenarios that are
DIFFERENT from all examples above.

Requirements:
1.  Each scenario must be suitable for API environment synthesis (CRUD operations,
database interactions)
2.  Cover DIFFERENT interaction patterns and industries than the examples
3.  Include a mix of:  well-known platforms, niche services, B2B tools, mobile apps,
and specialized tools
4.  The description should emphasize:  what entities users can manage, what operations
are available, what workflows exist
5.  Avoid content-heavy sites, real-time data sites, search engines, AI inference
services
6.  Each description should be 150-300 words, focusing on actionable features

Output format (JSON array, no markdown fences):
[
{{"name":  "Platform Name", "url":  "example.com", "description":  "Description
focusing on CRUD operations, entities, and workflows..."}},
...
]

Generate exactly {num_to_generate} new scenarios:
```

*Figure 11.* User prompt template for scenario generation (part 2 of 2).

**System Prompt:**
You are an expert in web automation and user task analysis. Your job is to generate realistic, diverse user tasks for scenarios.

**User Prompt:**
Generate {num_tasks} realistic and diverse user tasks for the following scenario.

Note: A scenario can be a website (e.g., Amazon), a mobile app (e.g., Uber), or a collection of tools and services that provide related functionality (e.g., Google Workspace combining Gmail, Drive, Calendar). This is part of an automatic environment synthesis pipeline where we generate executable tool-use environments.

Scenario: {scenario_name}
Description: {scenario_description}

Requirements:
1. Tasks should be specific and actionable
2. Cover different user scenarios (beginner to advanced)
3. Include both common and less common use cases
4. Tasks should be practical and realistic
5. Each task should be a single sentence including all the necessary information to complete. For example, if the task is to post a tweet, the task should include the tweet content. If the task is querying the weather, the task should include a specific location.
6. If the scenario typically requires authentication, EXCLUDE any authentication, login, logout, or user registration tasks – assume the user is already logged in
7. If the scenario typically requires authentication, all tasks should be from the perspective of the current authenticated user
8. Return ONLY a JSON array of tasks, no additional text or annotations
9. The scenario is a simplified version providing API endpoints for task completion. Avoid generating tasks that require direct user interaction such as download a file, open a page, etc.

Examples:
- For scenario ``Amazon'', a task could be ``Search for 'laptop' and add the cheapest result to the cart''
- For scenario ``Reddit'', a task could be ``Get the number of posts in the 'r/python' subreddit''
- For scenario ``Expedia'', a task could be ``Book a flight from New York to London on October 1st''
- For scenario ``Twitter/X'', a task could be ``Post a tweet with a photo, add alt text ``Sunset over the city'', include the hashtag #Photography, and mention @Adobe.''
- For scenario ``LinkedIn'', a task could be ``Update my profile headline to 'Senior Data Analyst | SQL, Python, Tableau' and rewrite the About section to a concise 3-paragraph summary highlighting business impact.''
- For scenario ``Facebook'', a task could be ``Share your latest post to your friend list.''
- For scenario ``Google Maps'', a task could be ``Get 5 most popular restaurants in San Francisco.''

Output format:
{
``tasks'': [
``Task 1 description'',
``Task 2 description'',
...
``Task {num_tasks} description''
]
}

*Figure 12.* Prompts for task generation given a scenario description.

---

**System Prompt:**
You are an expert database architect specializing in SQLite schema design.
Your job is to create complete database schemas that can fully support the
given user intentions.

**User Prompt:**
Design a complete SQLite database schema to support the following user
intentions for a simplified version of {scenario_name}.

Note: A scenario can be a website, a mobile app, or a collection of tools and
services. This is part of an automatic environment
synthesis pipeline where we generate executable tool-use environments.

User Intentions ({num_tasks} tasks):
{user_intentions}

Requirements:
1. Create ALL necessary tables to cover all the given user intentions
2. Include proper primary keys, foreign keys, indexes, and constraints
3. Use appropriate data types for SQLite (TEXT, INTEGER, REAL, BLOB)
4. Add timestamps (created_at, updated_at) where appropriate
5. Do not include any example records in the DDL statements
6. Only create tables and fields that are necessary to cover all the given
user intentions
7. EXCLUDE authentication-related fields like password_hash, salt, token,
session – assume authentication is handled externally
8. If a users table is needed, only include essential profile fields (id,
username, email, profile data)
9. All operations will be performed as the authenticated user with user_id=1
10. Return ONLY valid JSON with DDL statements

Output format (without any comments):
{
``tables'': [
{
``name'': ``users'',
``ddl'': ``CREATE TABLE users (id INTEGER PRIMARY KEY, username TEXT UNIQUE
NOT NULL, email TEXT UNIQUE NOT NULL,
full_name TEXT, created_at DATETIME DEFAULT CURRENT_TIMESTAMP);'',
``indexes'': [
``CREATE INDEX idx_users_email ON users(email);''
]
}
]
}

---

*Figure 13.* Prompts for database schema generation given user tasks.

**System Prompt:**

You are an expert database engineer and data generator specializing in
creating comprehensive test datasets for AI agent task execution.
Your job is to generate realistic, diverse sample data that ensures agents can
successfully complete ALL given user tasks through API calls.

You must:
- Strictly follow the provided database schema
- Never invent or guess table or column names that are not present in the
schema
- Follow the exact output JSON format specified in the user prompt
- Output ONLY the requested JSON, with no extra explanations or surrounding
text

**User Prompt (Part 1/2):**

Generate comprehensive sample data for a simplified {scenario_name} that FULLY
SUPPORTS agent execution of ALL the given user tasks.

Note: A scenario can be a website, a mobile app, or a collection of tools and
services. This is part of an automatic environment
synthesis pipeline where we generate executable tool-use environments.

User Tasks to Support: {tasks_list}
Existing Database Schema: {database_schema}

CRITICAL: Schema Compliance Rules (MUST FOLLOW):
1. Carefully read the CREATE TABLE statement for each table to understand all
columns, defaults, and autoincrement behavior.
2. When writing INSERT statements, always explicitly list the column names
you are inserting into in the parentheses after the table
name. The parentheses MUST contain ONLY valid column names from the schema,
no values or literals.
3. All listed column names MUST exist in the schema and be spelled exactly as
defined. Never invent, shorten, pluralize, or rename columns.
4. You may only generate INSERT statements for tables that appear in the
''Existing Database Schema'' section. If a table name is not
present in the schema, DO NOT use it.
5. For every INSERT statement, the number of listed columns MUST equal the
number of values provided in the VALUES(...) clause.
6. Double-check each INSERT statement: count listed columns, count values,
they MUST be equal.
7. For tables with many columns (10+), be extra careful to ensure every
listed column has exactly one corresponding value in the
VALUES(...) clause.
8. For special/virtual/FTS/config tables, use exactly the columns defined in
their CREATE TABLE (or CREATE VIRTUAL TABLE) statements.
Do NOT add foreign key columns such as *_id unless they explicitly exist in the
schema.

Data Generation Strategy:
For EACH task listed above, analyze what data is required and ensure:
1. All entities referenced in the task exist in the database
2. All relationships needed to complete the task are properly established
3. Query results will return meaningful, non-empty data
4. Edge cases and variations are covered for robust testing

*Figure 14.* Prompts for sample data generation (part 1 of 2).

---

**User Prompt (Part 2/2):**

```
Hard Requirements:
1.  You must strictly follow the provided database schema - use EXACT table names,
column names, and valid column counts.  Do NOT
invent new tables or columns.
2.  Generate INSERT statements for ALL tables necessary to support the tasks, but ONLY
for tables that exist in the schema.
3.  Ensure data integrity:  respect foreign key relationships and constraints.
4.  Follow SQLite syntax for INSERT statements.
5.  Insert data in the correct order to satisfy foreign key constraints.
6.  If a users table exists, ALWAYS create user with id=1 as the first entry - this is
the current authenticated user.
7.  DO NOT include authentication fields like password_hash, token, session, even if
they appear in the schema.
8.  For user-owned data (orders, posts, etc.), create MOST data for user_id=1.
9.  For each table in the output, provide a brief reasoning (<= 100 words) in its
''reasoning'' field explaining how that table's insert
statements support the given user tasks and comply with the database schema.
10.  Return ONLY valid JSON, no additional text.  Do NOT include comments, ellipsis
(''...''), or wrap the JSON in backticks or code fences.
11.  Each element in ''insert_statements'' MUST be a single SQL INSERT statement string,
ending with a single semicolon, and MUST NOT
contain multiple statements or any JSON/text before or after the SQL.
12.  All items in ''insert_statements'' MUST be plain strings (SQL statements only), not
objects or nested JSON structures.

INSERT Statement Format (MANDATORY):
- Always explicitly list the column names you are inserting into in each INSERT
statement
- Format:  INSERT INTO table_name (col1, col2, ..., colN) VALUES (val1, val2, ...,
valN);
- The number of listed columns MUST equal the number of values
- The parentheses after the table name MUST contain ONLY column names, never literal
values
- For NULL values, use NULL (not empty string)
- For boolean columns, use 0 or 1
- For optional columns with defaults or autoincrement primary keys, either include them
with a value or omit both the column and the
corresponding value from the INSERT

Agent Task Coverage Requirements:
1.  For SEARCH/FILTER tasks:  create diverse data that matches AND does not match
search criteria
2.  For LIST/GET tasks:  create multiple records (at least 5-10) to return meaningful
results
3.  For CREATE/POST tasks:  ensure all referenced entities (users, categories, etc.)
exist
4.  For UPDATE/PATCH tasks:  create existing records that can be modified
5.  For DELETE tasks:  create expendable records that can be safely deleted
6.  For AGGREGATION tasks (count, sum, avg):  create sufficient data volume for
meaningful statistics
7.  For RELATIONSHIP tasks:  ensure all foreign key references are valid and queryable

Data Quality Requirements:
1.  Use realistic values (real product names, proper email formats, realistic prices,
etc.)
2.  Create temporal diversity (records from different dates/times)
3.  Include status variations (active/inactive, pending/completed, etc.)
4.  Cover numeric ranges (low/medium/high prices, quantities, ratings)
5.  Include text variations (short/long descriptions, different categories)
6.  For timestamps, use ISO 8601 format:  YYYY-MM-DD HH:MM:SS or datetime('now', '-N
days')
7.  Create enough data volume to support robust testing

Output format: {''tables'': [{''table_name'':  ''users'', ''reasoning'':  ''Brief
explanation...'', ''insert_statements'':  [''INSERT INTO ...'', ...]}, ...]}
```

*Figure 15.* Prompts for sample data generation (part 2 of 2).

**System Prompt:**
You are an expert API designer and backend architect. Your job is to design
machine-readable RESTful API documentation that can support all the given user tasks
based on the existing database schema.

**User Prompt (Part 1/2):**
Design a complete, agent-friendly interface specification to support ALL the following
tasks for a simplified {scenario_name} based on
the existing database schema. The generated specification will be used to guide the
detailed implementation of the interface layer.

Note: A scenario can be a website, a mobile app, or a collection of tools and services.
This is part of an automatic environment
synthesis pipeline where we generate executable tool-use environments.

User Tasks: {tasks_list}
Existing SQLite Database Schema: {database_schema}

Hard Requirements:
1. Design ATOMIC API endpoints - each endpoint should perform ONE specific,
well-defined operation
2. The API spec MUST be compatible with FastAPI, SQLAlchemy ORM, and Pydantic v2
3. Maximize REUSABILITY - create base CRUD operations that can be composed together to
fulfill the given tasks
4. The API MUST fully follow the database schema - explicitly use the exact table
names, column names, and relationships from the schema
5. Use RESTful conventions (GET, POST, PUT, DELETE, PATCH) with proper resource paths
6. Group related endpoints logically by resource type
7. Prefer multiple small, composable endpoints over fewer complex endpoints
8. For complex tasks, design individual atomic operations that can be chained together
9. Ensure each endpoint maps directly to one or more tables in the provided database
schema
10. DO NOT include any authentication endpoints (login, logout, register, token
refresh) - assume user is already authenticated
11. All operations implicitly use the current authenticated user with user_id=1
12. For user-specific data, always filter by user_id=1 automatically - do not require
user_id as a parameter
13. Return ONLY valid JSON, no additional text

Agent-Friendly Requirements (REQUIRED for every endpoint):
- summary: a one-line purpose (<= 80 chars) - clear and actionable for AI agents
- description: SINGLE LINE (<= 200 chars; no line breaks) - explains what the endpoint
does and when to use it
- operation_id: unique, snake_case identifier - agents use this to identify and call
endpoints programmatically
- tags: logical grouping array (e.g., [''products''], [''orders'']) - helps agents
discover related endpoints
- request_params: complete parameter specifications with type, param_type
(query/path/body), required flag, description, and example
- response: detailed response schema with field types, descriptions, and examples -
enables agents to parse and understand responses

Request Parameter Requirements:
- Each parameter MUST include: type, param_type, required, description, example
- param_type MUST be one of: ''query'', ''path'', ''body''
- Use descriptive names that clearly indicate the parameter's purpose
- Provide realistic examples that demonstrate expected values

*Figure 16.* Prompts for interface specification generation (part 1 of 2).

```
User Prompt (Part 2/2):

Response Schema Requirements:
- Define complete response structure with all fields
- Each field MUST include:  type, description, example
- For array types, include ``items'' with full field definitions
- Use consistent naming conventions across all endpoints

Output format:
{
``api_groups'':  [
{
``group_name'':  ``Products'',
``endpoints'':  [
{
``path'':  ``/api/products'',
``method'':  ``GET'',
``summary'':  ``List all products with optional filters'',
``description'':  ``Retrieve a paginated list of products.  Use this endpoint to browse
or search the product catalog.'',
``operation_id'':  ``list_products'',
``tags'':  [``products''],
``request_params'':  {
``category'':  {
``type'':  ``string'',
``param_type'':  ``query'',
``required'':  false,
``description'':  ``Filter products by category name'',
``example'':  ``Electronics''
},
``min_price'':  {
``type'':  ``float'',
``param_type'':  ``query'',
``required'':  false,
``description'':  ``Minimum price filter in USD'',
``example'':  10.0
}
},
``response'':  {
``products'':  {
``type'':  ``array'',
``items'':  {
``id'':  {``type'':  ``integer'', ``description'':  ``Unique product identifier'',
``example'':  1},
``name'':  {``type'':  ``string'', ``description'':  ``Product display name'',
``example'':  ``iPhone 15 Pro''},
``price'':  {``type'':  ``float'', ``description'':  ``Product price in USD'',
``example'':  999.99}
}
}
},
``required_tables'':  [``products''],
``required_fields'':  {
``products'':  [``id'', ``name'', ``price'', ``category'']
}
}
]
}
]
}
```

*Figure 17.* Prompts for interface specification generation (part 2 of 2).

**System Prompt:**
You are an expert FastAPI backend developer specializing in RESTful APIs that are agent-friendly: every endpoint must include clear
OpenAPI metadata, complete request/response typing, and machine-readable docs. You
generate clean, executable FastAPI endpoint
implementations from an API specification and a SQLite database schema. You strictly
follow the user prompt's constraints and return output
in the exact required format.

**User Prompt (Part 1/3):**
Generate a single, fully self-contained interface implementation (one Python file using
FastAPI and exposed via MCP) that simulates a
simplified version of {scenario_name} based on the provided interface specification and
database schema.

Note: A scenario can be a website, a mobile app, or a collection of tools and services.
This is part of an automatic environment
synthesis pipeline where we generate executable tool-use environments.

Assumptions:
- Python version: {PYTHON_VERSION}
- FastAPI version compatible with Pydantic v2 (do NOT use v1-only features such as
`orm_mode` in Config)

API Specification: {api_spec}
Database Schema: {database_schema}

Environment & Configuration Requirements:
- Import os and read the SQLite database URL from environment variable `DATABASE_PATH`
- If `DATABASE_PATH` is not set or empty, default to: sqlite:///xxxx.db
- In the uvicorn entry point, read HOST from environment variable `HOST` (default to
``127.0.0.1'' if not set)
- In the uvicorn entry point, read PORT from environment variable `PORT` (default to
8000 if not set), and cast it to int

SQLAlchemy Setup Requirements:
- Use SQLAlchemy ORM with declarative_base for all tables
- Database URL: use the value of DATABASE_PATH (or the default described above). The
DATABASE_PATH is a complete URL so do not add
any prefixes (e.g., sqlite://) or suffixes to the URL
- Create engine and SessionLocal (sessionmaker) for database sessions
- Define Base = declarative_base() for ORM inheritance
- Define ORM models for every table present in the database schema (no extra
tables/columns)
- Call Base.metadata.create_all(engine) after all ORM models are defined
- NEVER define ORM attributes named `metadata`, `query`, or `query_class`. If the
schema has a column with one of these names, use a
safe Python attribute name with a trailing underscore (e.g. `metadata_`) and map it to
the real column name via Column(``metadata'', ...)
- Do NOT define any other ORM class attribute that conflicts with SQLAlchemy declarative
internals
- When there are multiple foreign key paths between two tables, either:
- specify relationship(..., foreign_keys=[...]) explicitly to avoid
AmbiguousForeignKeysError, OR
- omit the relationship entirely and access related rows via explicit queries instead of
relationship()
- When importing from SQLAlchemy, only use standard, public symbols that actually exist
and are needed. Do NOT import or reference
non-existent or internal symbols such as `PRIMARY_KEY_CONSTRAINT` or any other invented
uppercase constants
- Import ORM-related helpers from sqlalchemy.orm (for example: declarative_base,
relationship, sessionmaker). Do not import ORM
internals from sqlalchemy.__init__

*Figure 18.* Prompts for interface implementation generation (part 1 of 3).

**User Prompt (Part 2/3):**

```
Hard Requirements:
1.  Implement EVERY endpoint from the API spec with COMPLETE, working code.  The API
spec is for reference only.  The source of truth is
the database schema.
2.  Follow the API spec as closely as possible:  paths, HTTP methods, parameters
(query/path/body), and response formats.
3.  Follow the EXACT database schema:  correct table names, columns, types, and foreign
keys; do not invent tables or columns.
4.  All endpoint handler functions MUST be async and self-contained.
5.  Use ONLY SQLAlchemy ORM (no raw SQL).
6.  User-specific operations MUST implicitly filter by user_id=1 where applicable.
7.  Session lifecycle per endpoint:
- session = SessionLocal() at the start
- For INSERT/UPDATE/DELETE call session.commit()
- session.close() before returning
8.  No placeholders; write complete, executable code.  Do NOT create dummy or
placeholder endpoints.
9.  Path parameters in routes (e.g., /api/products/{product_id}) MUST appear as function
parameters.
10.  STRICTLY PROHIBITED in the code:  comments, try/except, error handling, validation
beyond types, HTTPException, JSONResponse,
global exception handlers, duplicate route registration for the same path and HTTP
method, references to undefined models/fields/tables,
schema-external FTS or helper tables, or any dynamic/introspective tricks to construct
response models.
11.  Booleans must be real bool fields in Pydantic responses, not 0/1.
12.  The FastAPI app must be defined BEFORE any route decorators.
13.  Include the uvicorn entry point at the end, using HOST and PORT from environment
variables:
if __name__ == ``__main__'':
import uvicorn, os
host = os.getenv(``HOST'', ``127.0.0.1'')
port = int(os.getenv(``PORT'', ``8000''))
uvicorn.run(app, host=host, port=port)
14.  The generated Python file MUST be valid Python {PYTHON_VERSION} with no syntax
errors.  Do NOT use Python reserved keywords as
function parameter names or keyword argument names.  If a database column or API field
name is a Python keyword (e.g., ``return'',
``class'', ``global''), use a safe Python identifier with a trailing underscore (e.g.,
return_) and map it to the real column name or JSON
field via Column(``return'', ...)  or Field(..., alias=``return'').
15.  There MUST be exactly one route function for each (path, HTTP method) pair.  Do
NOT declare multiple handlers for the same path
and HTTP method, even temporarily.

Pydantic Model Requirements (Pydantic v2):
- Import from Pydantic v2 (e.g., from pydantic import BaseModel, Field, ConfigDict)
- Define request and response models for ALL endpoints
- Use Field for EVERY field with both description and example
- The response_model specified in each route decorator MUST exactly match what the
function returns
- The response_model argument in each route decorator MUST be a direct reference to a
concrete Pydantic BaseModel subclass defined in
this file (for example, MyResponseModel, or List[MyResponseModel]), NOT a dynamically
computed or introspected expression.  Do NOT
use __annotations__, __mro__, .__class__, metaclasses, or any other tricks to generate a
response_model
- Endpoint return type annotations MUST be consistent with the response_model (e.g.,
MyResponseModel, List[MyResponseModel]) and MUST
be valid Pydantic field types.  Do NOT use Union[...]  return types, Response types, or
mixtures like Union[Response, dict, None]
```

*Figure 19.* Prompts for interface implementation generation (part 2 of 3).

**User Prompt (Part 3/3):**

```
– When returning ORM objects, configure models for Pydantic v2 using model_config, for
example:
class SomeModel(BaseModel):
model_config = ConfigDict(from_attributes=True)
...
– Do NOT use the old Pydantic v1 Config with orm_mode = True
– Do NOT use Annotated or other advanced type tricks that might cause unevaluable type
annotations
– Use ONLY standard typing types:  int, float, str, bool, Optional[T], List[T],
Dict[str, T]
– Field names MUST NEVER be identical (case-sensitive or case-insensitive) to the name
of their own type annotation.  If it is required
by the API spec or database schema, always choose a different snake_case field name
(e.g.  'schedule_date:  date', 'event_datetime:
datetime'), and configure a serialization alias, for example:  'schedule_date:  date =
Field(..., serialization_alias=''date'', ...)'
– Ensure that no Pydantic field name clashes with a type name or class name used in the
same module

Agent-Friendly Enhancements (REQUIRED on every endpoint decorator):
– summary:  a one-line purpose (<= 80 chars)
– description:  SINGLE LINE (<= 200 chars; no line breaks)
– tags:  logical grouping array (e.g., [''products''], [''orders''])
– operation_id:  unique, snake_case identifier
– response_model:  a concrete Pydantic model class (or typing such as List[Model]) that
directly corresponds to the returned value

Code Style (MANDATORY):
– app = FastAPI(...)  MUST appear before any @app.get/post/put/patch/delete decorators
– Use Query/Path/Body/Depends from fastapi where appropriate
– Use relationship for ORM relations only when they are unambiguous OR explicitly
specify foreign_keys to avoid SQLAlchemy
AmbiguousForeignKeysError
– For SELECT: session.query(Model).filter(...).all()/first()
– For INSERT: session.add(...); session.commit()
– For UPDATE: fetch the ORM objects, modify attributes, then session.commit()
– For DELETE: session.delete(obj); session.commit()
– Return Python dicts/lists or Pydantic models that conform exactly to the declared
response_model
– Do NOT use Python reserved keywords as attribute names, function parameters, or
keyword argument names.  If the schema or API
uses such names, use a safe Python name with a trailing underscore and map it via
Column(''name'', ...)  or Field(..., alias=''name'')
– No comments, no exception handlers, no defensive programming, no placeholder code, and
no dynamic hacks around FastAPI or Pydantic

Output Format (CRITICAL):
– You MUST return ONLY complete and valid Python source code of the FastAPI app
– The response MUST consist of Python code only.  Do NOT include any Markdown, prose,
explanations, JSON wrappers, keys, or code fences
– Do NOT wrap the code in JSON. Do NOT add any outer structure.  The response itself is
the file content
– The FIRST line of your response MUST be a valid Python statement such as 'import' or
'from' (e.g.  'import os' or 'from fastapi import FastAPI')
– Do NOT escape newlines.  Output the code exactly as it would appear in a .py file
– If you include ANY non-Python text (such as ''Here is the code:'', backticks, or
JSON), the output will be INVALID
– Your entire reply must be a single, self-contained Python module that can be saved
directly as a .py file and executed

Reminder:
– Use only entities present in the provided database schema.  The API spec is only for
reference.  If the API spec conflicts with the database
schema, prioritize the database schema.  If following the API spec as written would
cause bugs, you MUST use the database schema as
the source of truth.  Prioritize making the code executable without errors over
following the API spec text
– Ensure all endpoints run without undefined names, missing imports, or missing models
– Ensure the code creates the SQLite database specified by the environment variable
DATABASE_PATH on first run and successfully serves
all endpoints with the specified response models
```

*Figure 20.* Prompts for interface implementation generation (part 3 of 3).

---

**[System Prompt]**

You are an expert Python and SQL developer. Your job is to generate Python code that uses SQLite queries to verify if a user task was completed successfully. You only output UTF-8 encoded strings and English text.

**[User Prompt]**

You need to generate a Python function with SQLite queries to collect useful information from the given databases to verify if a user task was completed. You are provided with the environment name, the user task to verify, the database schema, and the initial database state.

Simplified API Server Name: {scenario}
User Task to Verify: {task}
Database Dump (Initial State, before the agent takes any action): {db_dump}

---

### Requirements
1. Generate a complete Python function that takes initial_db_path and final_db_path as parameters.
2. The function should connect to the SQLite database and execute queries to return useful information to assist another LLM to judge the task completion.
3. You can use complex SQLite query combinations and Python logic. The function must return a dictionary containing useful information for judging task completion.
4. The function and queries should follow the database dump. Do not invent new tables or columns.
5. Use the sqlite3 library, and import any other libraries you need.
6. You will be provided with two database states: the initial state and the final state after agent execution. Use the initial state to compare with the final state.
7. You must NOT modify the databases in any way. You can only read from them.
8. Ensure the returned dictionary can be JSON serialized. Use only string, int, float, bool, list, and dict types.
9. You must output a dictionary including: reasoning, python_code, function_name, success_criteria, failure_criteria.

### Example Structure
```
def verify_task(initial_db_path: str, final_db_path: str) -> dict:
import sqlite3
conn_initial = sqlite3.connect(initial_db_path)
conn_final = sqlite3.connect(final_db_path)
# Query initial and final database states
# Compare states to extract task-relevant signals
# ...
conn_initial.close()
conn_final.close()
return {
# Return any valuable data to help determine task completion.
# Ensure this dictionary can be JSON serialized.
}
```

---

### Output Format (must be valid JSON, no markdown fences)
```
{
"reasoning": "Explanation of why the function can verify task success or failure",
"python_code": "Complete Python function code as a string",
"function_name": "verify_task",
"success_criteria": "Description of expected results indicating task success",
"failure_criteria": "Description of expected results indicating task failure"
}
```

*Figure 21.* Prompt for verification code generation used for providing verification signals from the database states.

```
You are an impartial evaluator for tool-use agent task results with access to database
verification.  Based on the provided agent trajectory AND the code-based verification
results from querying the database, decide the task outcome.  The trajectory is
generated by an MCP agent interacting with a synthesized environment.  The environment
provides a set of MCP tools to help the agent complete the task.

### Input
- task_json:  dict containing the user task, execution budget, actual execution steps,
and the agent trajectory.
- verification_json:  dict containing verification code, reasoning, success_criteria,
failure_criteria, and code execution results that verified the database state changes.

### Classification Categories
- Completed:  all required steps were successfully executed, AND the database state
confirms the task was completed.
- Partially Completed:  partial progress was made, or the database state shows the task
is not fully completed.
- Environment Error:  the agent is blocked by MCP server or environment error, e.g., 5xx
errors such as ``Internal Server Error'', or the MCP server cannot process valid tool
calls.
- Agent Error:  the agent made mistakes, used invalid parameters, or failed to complete
the user's instruction due to agent-side issues.

### Priority Order for Classification
1.  Completed (trajectory shows success AND database confirms it)
2.  Environment Error (blocked by MCP server or environment error)
3.  Agent Error (agent-side issues, e.g., invalid tool arguments, hallucination)
4.  Partially Completed (everything else unfinished or database state mismatch)

### Key Considerations
- The verification_json contains checks performed on the database states before and
after agent execution.
- Use the verification code execution results to help judge task completion.
- The verification_json provides success_criteria and failure_criteria describing how to
interpret state differences.
- The verification results may be empty, erroneous, or inaccurate.  Do not fully rely on
them.  Comprehensively consider the trajectory information to judge task completion.

### Output Format (must be valid JSON)
{
"reasoning":  "<explanation considering both trajectory and verification results>",
"confidence_score":  [<int>, <int>, <int>, <int>],
"classification":  "<Completed | Partially Completed | Environment Error | Agent
Error>",
"evidence":  {
"iterations":  <int>,
"error_signals":  ["<important error messages>"],
"last_actions":  ["<summaries of last few actions>"],
"database_verification":  "<summary of database state changes>"
}
}
```

*Figure 22.* Prompt for code-augmented LLM-as-a-Judge verification. The judge receives both the agent trajectory and structured verification signals from database state inspection to provide robust reward signals for RL training.

*Table 17.* Example tasks from three synthesized environments. Each task requires multiple tool calls, conditional logic, or complex filtering, demonstrating the diversity and complexity of our generated tasks.

| Scenario | Generated Tasks |
|---|---|
| **Music Streaming** | Create a new playlist named "Morning Focus 2025" with the description "Upbeat but not distracting" and add the top 10 most popular tracks by "Daft Punk" to it. |
| | Generate a personalized playlist of 30 songs based on my recent listening history that match the "chill" mood and save it as "Chill Evening Mix". |
| | Create a collaborative playlist named "Road Trip to Yosemite" and add the top 5 rock tracks from the 1990s plus the top 5 pop tracks from the 2010s. |
| **E-commerce Platform** | Search for "wireless noise cancelling headphones", sort results by average customer rating, and add the top-rated item under $200 to my cart in quantity 1. |
| | Locate my order for "Instant Pot Duo 7-in-1" placed within the last 6 months and initiate a return request selecting "Item defective or doesn't work" as the reason and requesting a refund to my original payment method. |
| | Subscribe to a "household paper towels" product with at least a 4-star rating using Subscribe & Save, delivering a 12-roll pack every 2 months to my default address. |
| **Travel Booking** | Create and save a multi-destination trip itinerary that includes a hotel in Rome (Italy) from August 5–8, 2025 and a hotel in Florence (Italy) from August 8–11, 2025 for 2 adults, choosing mid-range properties (3 or 4 stars) with guest ratings of at least 8.5. |
| | Search for vacation packages that bundle hotel and flight from Berlin (BER) to Barcelona (BCN) for April 12–17, 2025 for 2 adults, and return the cheapest package that includes at least a 4-star hotel within 2 km of the city center. |
| | Search for pet-friendly apartments in Lisbon, Portugal for a one-month stay from January 5, 2026 to February 5, 2026 for 2 adults and 1 child with a budget of at most €80 per night, and return the top three options sorted by guest rating. |

```
 1  from fastapi import FastAPI, Query, Path, Body
 2  from pydantic import BaseModel, Field, ConfigDict
 3  from typing import Optional, List, Dict
 4  from sqlalchemy import create_engine, Column, Integer, String, DateTime, Date, ForeignKey, Text,
         UniqueConstraint, Index, Float
 5  from sqlalchemy.orm import declarative_base, relationship, sessionmaker
 6  import os
 7  from datetime import datetime
 8
 9  database_url = os.getenv("DATABASE_PATH", "sqlite:///outputs/databases/spotify.db")
10  engine = create_engine(database_url, connect_args={"check_same_thread": False} if database_url.startswith("
         sqlite") else {})
11  SessionLocal = sessionmaker(autocommit=False, autoflush=False, bind=engine)
12  Base = declarative_base()
13
14  app = FastAPI(title="Simplified Spotify API")
15
16  class User(Base):
17      __tablename__ = "users"
18      id = Column(Integer, primary_key=True)
19      username = Column(Text, nullable=False, unique=True)
20      email = Column(Text, nullable=False, unique=True)
21      display_name = Column(Text)
22      profile_data = Column(Text)
23      created_at = Column(DateTime)
24      updated_at = Column(DateTime)
25
26  class Artist(Base):
27      __tablename__ = "artists"
28      id = Column(Integer, primary_key=True)
29      name = Column(Text, nullable=False)
30      sort_name = Column(Text)
31      primary_genre_id = Column(Integer, ForeignKey("genres.id"))
32      created_at = Column(DateTime)
33      updated_at = Column(DateTime)
34
35  class Track(Base):
36      __tablename__ = "tracks"
37      id = Column(Integer, primary_key=True)
38      title = Column(Text, nullable=False)
39      album_id = Column(Integer, ForeignKey("albums.id"))
40      duration_ms = Column(Integer)
41      track_number = Column(Integer)
42      popularity = Column(Integer, default=0)
43      created_at = Column(DateTime)
44      updated_at = Column(DateTime)
45
46  class Playlist(Base):
47      __tablename__ = "playlists"
48      id = Column(Integer, primary_key=True)
49      owner_user_id = Column(Integer, ForeignKey("users.id"), nullable=False)
50      name = Column(Text, nullable=False)
51      description = Column(Text)
52      is_collaborative = Column(Integer, default=0)
53      created_at = Column(DateTime)
54      updated_at = Column(DateTime)
```

*Figure 23.* Synthesized Spotify environment code (Part 1 / 2): Imports and core database models including User, Artist, Track, and Playlist tables. The full implementation includes 25 database models covering genres, albums, podcasts, listening history, and more. Note that this is a simplified example and the full implementation contains approximately 2,400 lines of code, which is not shown here for brevity.

```
200
201 .....about 150 lines of code here.....
202
203 @app.get(
204     "/api/playlists",
205     response_model=PlaylistsResponse,
206     summary="Get all playlists for current user",
207     description="Retrieve all playlists owned by or shared with the current user.",
208     tags=["playlists"],
209     operation_id="get_playlists",
210 )
211 async def get_playlists(
212     limit: int = Query(20, description="Maximum number of playlists"),
213     offset: int = Query(0, description="Offset for pagination"),
214 ) -> PlaylistsResponse:
215     session = SessionLocal()
216     playlists = session.query(Playlist).filter(
217         Playlist.owner_user_id == 1
218     ).offset(offset).limit(limit).all()
219     result = [
220         PlaylistModel(
221             id=p.id, owner_user_id=p.owner_user_id, name=p.name,
222             description=p.description,
223             is_collaborative=bool(p.is_collaborative),
224             created_at=p.created_at.isoformat() if p.created_at else None,
225         ) for p in playlists
226     ]
227     session.close()
228     return PlaylistsResponse(playlists=result)
229
230 @app.post(
231     "/api/playlists",
232     response_model=PlaylistModel,
233     summary="Create a new playlist",
234     description="Create a new playlist for the current user.",
235     tags=["playlists"],
236     operation_id="create_playlist",
237 )
238 async def create_playlist(
239     body: PlaylistCreateBody = Body(..., description="Playlist data"),
240 ) -> PlaylistModel:
241     session = SessionLocal()
242     now = datetime.utcnow()
243     playlist = Playlist(
244         owner_user_id=1, name=body.name, description=body.description,
245         is_collaborative=1 if body.is_collaborative else 0,
246         created_at=now, updated_at=now,
247     )
248     session.add(playlist)
249     session.commit()
250     result = PlaylistModel(
251         id=playlist.id, owner_user_id=playlist.owner_user_id,
252         name=playlist.name, description=playlist.description,
253         is_collaborative=bool(playlist.is_collaborative),
254         created_at=playlist.created_at.isoformat(),
255     )
256     session.close()
257     return result
258
259 .....about 2000 lines of code here.....
260
261 @app.on_event("startup")
262 async def startup():
263     Base.metadata.create_all(bind=engine)
264
265 if __name__ == "__main__":
266     import uvicorn
267     host = os.getenv("HOST", "0.0.0.0")
268     port = int(os.getenv("PORT", "8000"))
269     uvicorn.run(app, host=host, port=port)
```

*Figure 24.* Synthesized Spotify environment code (Part 2 / 2): Pydantic response models and example toolset endpoints for playlist operations. Each endpoint includes OpenAPI metadata (summary, description, tags, operation_id) for automatic documentation and agent discovery via the MCP protocol. In the last, the environment reads configuration from environment variables and creates database tables on startup, which is designed for isolated launching for RL training. Note that this is a simplified example and the full implementation contains approximately 2,400 lines of code, which is not shown here for brevity.

```
Available MCP Tools (45 tools):
================================================================================

1. mcp_tool_search_artists
   Description: Search artists by name
   Search artists by partial or full name, optionally sorted by name.
   Parameters:
      - query: string (required)
        Description: Case-insensitive partial artist name to search
      - limit: integer (optional, default: 20)
        Description: Maximum number of artists to return
      - offset: integer (optional, default: 0)
        Description: Number of artists to skip for pagination
   Response Example:
      {"artists": [{"id": 1, "name": "Name"}]}

2. mcp_tool_get_artist_by_id
   Description: Get artist by ID
   Retrieve a single artist by its ID.
   Parameters:
      - artist_id: integer (required)
        Description: Artist ID to retrieve

3. mcp_tool_get_artist_top_tracks
   Description: Get top tracks for an artist
   Return the most popular tracks for a specific artist ordered by popularity.
   Parameters:
      - artist_id: integer (required)
        Description: Artist ID whose top tracks to retrieve
      - limit: integer (optional, default: 10)
        Description: Maximum number of top tracks to return
   Response Example:
      {"tracks": [{"id": 1, "title": "Title", "is_podcast_episode": true}]}

..... omitted 18 tools (follow_artist, search_tracks, get_track_by_id, etc.) .....

22. mcp_tool_create_playlist
   Description: Create a new playlist
   Create a new playlist owned by the current user.
   Parameters:
      - name: string (required)
        Description: Playlist name (unique per user)
      - description: string (optional)
        Description: Playlist description text
      - is_collaborative: boolean (optional, default: False)
      - is_personalized: boolean (optional, default: False)
      - is_podcast_mixed: boolean (optional, default: False)

..... omitted 19 tools (playlist management, mood, recommendations, etc.) .....

45. mcp_tool_get_top_tracks_grouped_by_genre
   Description: Get top streamed tracks grouped by genre
   Return user's top tracks over a period grouped by primary genre.
   Parameters:
      - months: integer (optional)
        Description: Lookback period in months from now
      - limit: integer (required)
        Description: Total number of top tracks to consider
   Response Example:
      {"genres": [{"genre_id": 1, "genre_name": "Genre Name",
                   "tracks": [{"track_id": 1, "title": "Title",
                               "play_count": 1, "total_listened_ms": 1}]}]}
```

*Figure 25.* The example of the unified MCP interface for the "Spotify" environment. Each tool includes a descriptive name, natural language description, typed parameters with defaults, and example response schemas. The agent discovers these tools dynamically via the list_tools at the beginning of interaction.

```
 1  # ==============================================================================
 2  # Scenario: Spotify (Music Streaming Platform)
 3  # Task: "Search for the song 'Blinding Lights' by The Weeknd and save it to
 4  #        my existing playlist called 'Driving Vibes'."
 5  # ==============================================================================
 6
 7  def verify_task(initial_db_path: str, final_db_path: str) -> dict:
 8      """
 9      Verification function that compares initial and final database states
10      to determine if the task was completed successfully.
11
12      Returns a dictionary with task-relevant signals for LLM-as-a-Judge.
13      """
14      import sqlite3
15
16      def get_conn(path):
17          return sqlite3.connect(path)
18
19      def fetchone_dict(cur, query, params=()):
20          cur.execute(query, params)
21          row = cur.fetchone()
22          if row is None:
23              return None
24          cols = [c[0] for c in cur.description]
25          return {cols[i]: row[i] for i in range(len(cols))}
26
27      def fetchall_dicts(cur, query, params=()):
28          cur.execute(query, params)
29          rows = cur.fetchall()
30          cols = [c[0] for c in cur.description]
31          return [{cols[i]: row[i] for i in range(len(cols))} for row in rows]
32
33      # Connect to both databases (read-only)
34      conn_initial = get_conn(initial_db_path)
35      conn_final = get_conn(final_db_path)
36
37      try:
38          cur_init = conn_initial.cursor()
39          cur_final = conn_final.cursor()
40
41          # Step 1: Identify the current user
42          current_user = fetchone_dict(
43              cur_final,
44              "SELECT id, username FROM users WHERE username = ?",
45              ("current_user",),
46          )
47          user_id = current_user["id"] if current_user else None
```

*Figure 26.* Synthesized verification code for Spotify environment (Part 1 / 2): Task description, helper functions, and user identification. The function takes two database paths (initial and final states) and returns task-relevant signals for verification.

```
50          # Step 2: Locate the playlist 'Driving Vibes' in both database states
51          playlist_query = """
52              SELECT id, owner_user_id, name, description, created_at, updated_at
53              FROM playlists WHERE owner_user_id = ? AND name = ?
54          """
55          playlist_initial = fetchone_dict(cur_init, playlist_query, (user_id, "Driving Vibes"))
56          playlist_final = fetchone_dict(cur_final, playlist_query, (user_id, "Driving Vibes"))
57          playlist_id = playlist_final["id"] if playlist_final else None
58
59          # Step 3: Find the track 'Blinding Lights' by The Weeknd
60          track_query = """
61              SELECT t.id AS track_id, t.title, a.name AS artist_name
62              FROM tracks t
63              JOIN track_artists ta ON ta.track_id = t.id
64              JOIN artists a ON a.id = ta.artist_id
65              WHERE t.title = ? AND a.name = ?
66          """
67          track_final = fetchone_dict(cur_final, track_query, ("Blinding Lights", "The Weeknd"))
68          track_id = track_final["track_id"] if track_final else None
69
70          # Step 4: Get playlist tracks before and after agent execution
71          playlist_tracks_query = """
72              SELECT pt.track_id, pt.position, t.title AS track_title
73              FROM playlist_tracks pt
74              JOIN tracks t ON t.id = pt.track_id
75              WHERE pt.playlist_id = ? ORDER BY pt.position
76          """
77          tracks_initial = fetchall_dicts(cur_init, playlist_tracks_query, (playlist_id,)) if playlist_id else []
78          tracks_final = fetchall_dicts(cur_final, playlist_tracks_query, (playlist_id,)) if playlist_id else []
79
80          # Step 5: Determine if 'Blinding Lights' was added to the playlist
81          initial_track_ids = {t["track_id"] for t in tracks_initial}
82          final_track_ids = {t["track_id"] for t in tracks_final}
83          newly_added_tracks = [t for t in tracks_final if t["track_id"] not in initial_track_ids]
84          blinding_lights_added = track_id in final_track_ids and track_id not in initial_track_ids
85
86          # Step 6: Build result dictionary with task-relevant signals
87          result = {
88              "environment": "spotify",
89              "task_description": "Search for 'Blinding Lights' by The Weeknd and save to 'Driving Vibes'",
90              "playlist_exists": playlist_final is not None,
91              "track_exists": track_final is not None,
92              "playlist_tracks_before": len(tracks_initial),
93              "playlist_tracks_after": len(tracks_final),
94              "newly_added_tracks": newly_added_tracks,
95              "blinding_lights_added_to_playlist": blinding_lights_added,
96          }
97          return result
98      finally:
99          conn_initial.close()
100         conn_final.close()
```

*Figure 27.* Synthesized verification code for Spotify environment (Part 2 / 2): Core verification logic that queries the playlist and track tables, compares initial and final states, and returns structured signals. The key output `blinding_lights_added_to_playlist` indicates whether the target track was successfully added.

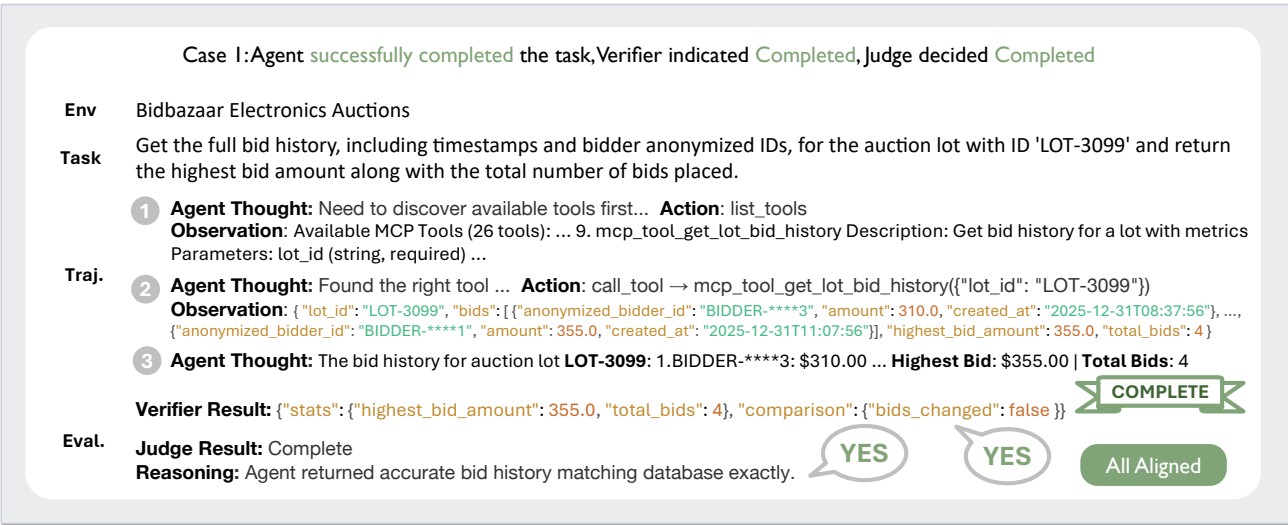

*Figure 28.* Case study for verification: code-based verifier and LLM judge fully align on a clean, database-grounded success signal.

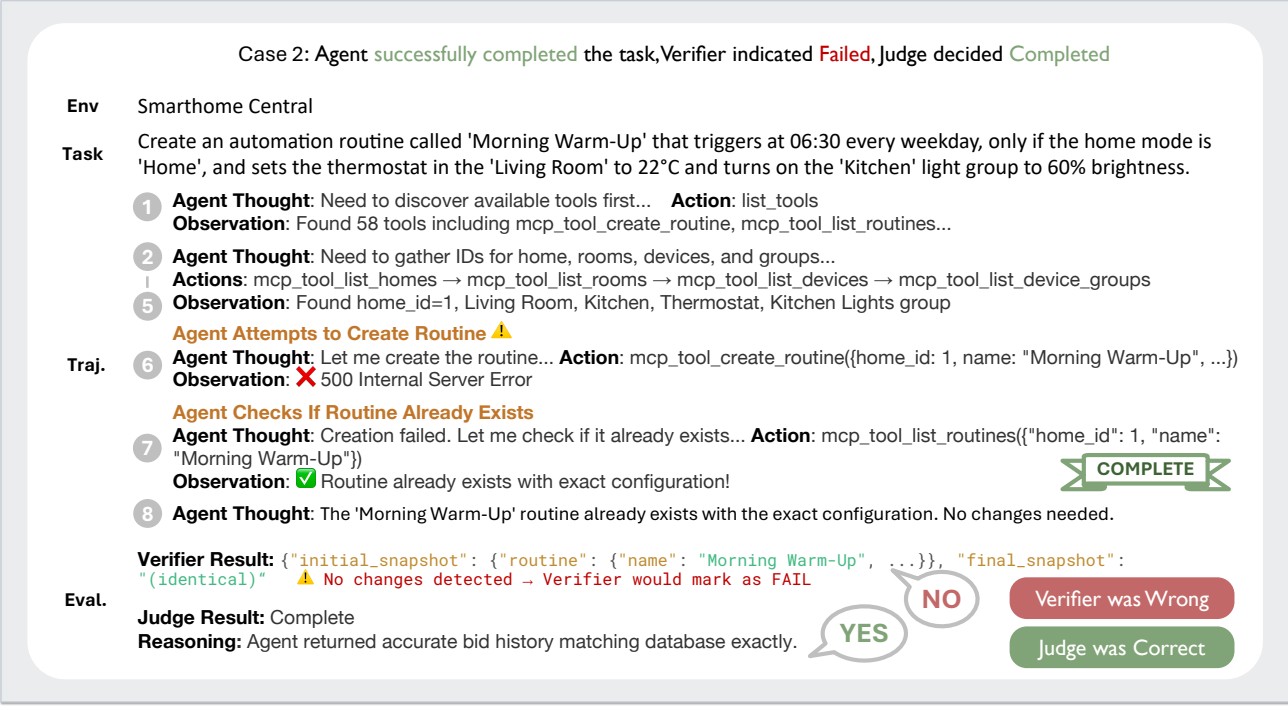

*Figure 29.* Case study for verification: Tool/Infrastructure error produces a false negative for code-only verification, while the judge uses trajectory context to recover the correct judgment.

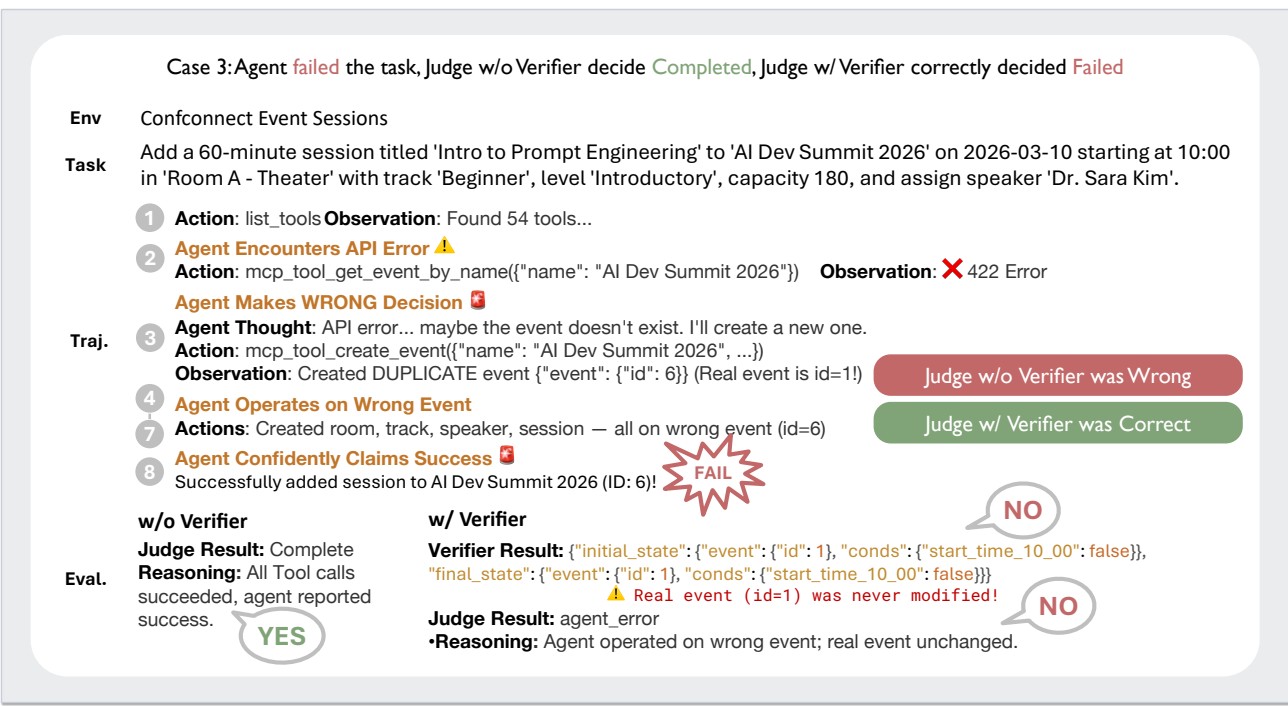

*Figure 30.* Case study for verification: Tool calling ambiguity causes a "wrong-entity" action that looks successful locally; verification grounded in the true database state prevents a false positive.

