# OpenReview forum: "Agent World Model: Infinity Synthetic Environments for Agentic Reinforcement Learning"
_ICML.cc/2026/Conference — ICML 2026 regular_

### Official Review · Reviewer_sNtv · 2026-03-12

**Soundness:** 4
**Presentation:** 4
**Significance:** 3
**Originality:** 2
**Overall Recommendation:** 5
**Confidence:** 5

**Summary:**

This paper introduces a scalable pipeline to generate synthetic environments for training and evaluation. The work also provides 1000 ready-made environments of everyday scenarios with an average of 35 tools each. Environments are for training and evaluating code agents (with tool use) and are backed by databases (for verification). Authors call their pipeline Agent World Model (AWM) that lets users generate environments at scale, adhering to MCP interfaces, verifiable by database states and includes self-correction. Domains of environments can be in the domain of "shopping, social media, finance, and travel".

**Compliance With Llm Reviewing Policy:**

Affirmed.

**Final Justification:**

All questions have been answered and concerns clarified, thank you.

**Key Questions For Authors:**

See Weaknesses & Questions.

**Limitations:**

See Weaknesses & Questions.

**Strengths And Weaknesses:**

### Strengths

- Very well written paper, really easy to understand and follow, great flow overall and structure
- Focus is on stateful application instead of static content for environment generation, this is very relevant and useful
- Authors provide thorough analysis through experimentation. This includes failure modes
- Diversity of generated is analyzed at scale, this is important and to be commended

### Weaknesses & Questions

- "In contrast to some works performing training on test environments," - Personally I haven't heared of that, but this needs reference/citation
- "However, because our environments are fully synthetic, verification can occasionally be affected by environment imperfections, such as incomplete state updates, unexpected execution failures, or infrastructurerelated issues (e.g., timeouts)." - How high is the failure percentage? What have your experiments shown, what do others experiments show?
- "The pipeline achieves over 85% success rates, and the self-correction mechanism requires only 1.13" - Does this relate to the point I made preceeding? If so, this needs to be tight together somehow.

---

> ### Author Rebuttal · Authors · 2026-03-30
>
> **Thank you for the strong endorsement and valuable feedback. We are happy to answer your questions as follows.**
>
> > Q1: "In contrast to some works performing training on test environments," - Personally I haven't heared of that, but this needs reference/citation
>
> We will add citations to works where agents are trained on benchmark-specific environments, such as:
> 1. using environment-specific tasks and data to train the simulator model [1,2]
> 2. direct training on/using benchmark-specific environments [3,4,5,6]
>
> Note that these works do not use the test task set, but they need to use the test environments either for interaction or for training.
>
> [1] Li, Yuetai, et al. "Simulating environments with reasoning models for agent training." arXiv preprint arXiv:2511.01824 (2025).
>
> [2] Chen, Zhaorun, et al. “Scaling Agent Learning via Experience Synthesis.” The Fourteenth International Conference on Learning Representations (ICLR), 2026
>
> [3] Luo, Michael. "Deepswe: Training a fully open-sourced, state-of-the-art coding agent by scaling rl, Jul 2025." Blog URL, https://www.together.ai/blog/deepswe
>
> [4] Prabhakar, Akshara, et al. "Apigen-mt: Agentic pipeline for multi-turn data generation via simulated agent-human interplay." arXiv preprint arXiv:2504.03601 (2025).
>
> [5] Wei, Zhepei, et al. "Webagent-r1: Training web agents via end-to-end multi-turn reinforcement learning." Proceedings of the 2025 Conference on Empirical Methods in Natural Language Processing. 2025.
>
> [6] Wang, Zihan, et al. "Ragen: Understanding self-evolution in llm agents via multi-turn reinforcement learning." arXiv preprint arXiv:2504.20073 (2025).
>
>
> > Q2: "However, because our environments are fully synthetic, verification can occasionally be affected by environment imperfections, such as incomplete state updates, unexpected execution failures, or infrastructurerelated issues (e.g., timeouts)." - How high is the failure percentage? What have your experiments shown, what do others experiments show?
>
> As mentioned in Sec. 6.1 (L370), the failure percentage is relatively low and stable, around 4%. Please see the anonymous figure link at [https://imgur.com/a/9MxcDnj](https://imgur.com/a/9MxcDnj) for the detailed failure percentage during RL training. We would like to add this into the revised version.
>
> Other concurrent works (e.g., EnvScaler & AutoForge), as far as we know, do not report the failure percentage for imperfect environments, nor do they design imperfect-environment rewards for RL training. However, according to our quality analysis on the generated environments of AWM & EnvScaler in Table 5, many tasks may be blocked due to environment errors in the code logic, indicating the necessity of designing such rewards for RL training.
>
>
> > Q3: "The pipeline achieves over 85% success rates, and the self-correction mechanism requires only 1.13" - Does this relate to the point I made preceeding? If so, this needs to be tight together somehow.
>
> Yes, they are connected. We would like to clarify the relationship between the pipeline success rate and the failure percentage for imperfect environments:
>
> 1. The success rate only reflects the ability of the pipeline to generate executable environments, but does not reflect the quality of the environments. For example, edge input cases may cause the environment to crash, which cannot be detected by simple execution tests.
> 2. In practice, we set an error threshold (10%) for the self-correction mechanism to handle generation errors. If the error rate is lower than the threshold (e.g., some statements can fail during database creation), we still accept the generation to reduce the costs, which can inevitably lead to some imperfect environments.
> 3. There are three numbers that describe different stages: synthesis (85% success rate) → quality assessment (74% contain bugs) → environment error rate during RL training (4%).
>
> We will restructure this narrative to make the progression clearer. Thank you for pointing out this gap.

---

> > ### Author Rebuttal · Reviewer_sNtv · 2026-04-07
> >
> > Thanks for your explanation.

---

> > > ### Author Response · Authors · 2026-04-07
> > >
> > > Thank you so much for your endorsement. We are happy to see our rebuttal has addressed your concerns!

---

### Official Review · Reviewer_Lzwt · 2026-03-13

**Soundness:** 3
**Presentation:** 3
**Significance:** 3
**Originality:** 2
**Overall Recommendation:** 4
**Confidence:** 4

**Summary:**

In this paper, the authors proposed a new pipeline of Agent World Model (AWM), aiming to synthesize many environments with tool coverage, code-driven, and backed by databases. The experiments resulted in 1000 environments, with around 10k tasks and 35k tools. The authors performed RL on tool calling agents with environments synthesized with their pipeline, and found that these trained agents achieve high performance on three benchmarks.

**Compliance With Llm Reviewing Policy:**

Affirmed.

**Final Justification:**

The rebuttal addressed my main concerns, so I maintained my positive scores.

**Key Questions For Authors:**

- Why is EnvScaler better than your models?
- Are the synthesized environments genuinely diverse?

**Limitations:**

Yes

**Strengths And Weaknesses:**

Summary of Strengths:
- The authors aim to address an important issue of a lack of diverse and executable environments.
- The pipeline the authors proposed is cost efficient, while able to provide relevantly diverse and useful environments at scale.
- The authors demonstrated the effectiveness of their synthesized environments through obtaining high performance on three benchmark with models trained with their synthesized environments.
- The authors performed comprehensive analysis such as quality analysis of synthesized environments, providing useful insights on potential issues with the synthesized environments.

Summary of Weaknesses:
- The authors only experimented on GPT5 for environment synthesis. It would be good if the authors could evaluated on another model family too, to see if the good results could generalize to other model families.
- The authors claimed that the results could generalize to OOD, yet the results are not as good as claimed, especially compared to EnvScaler. Additionally, the authors didn't explain on why EnvScaler is better than their models.
- Bug analysis reveals that 74% of the environments sampled contains bugs, with an average of more than 4 bugs per environment. The authors should add explanations to why the bugs are produced and what kind of bugs are there.
- The authors claimed that the synthesized environments are diverse, yet didn't show any quantitative analysis on the diversity of the synthesized environments. The authors should demonstrate that the synthesized environments differ genuinely, instead of just simple domain variations of each others.

---

> ### Author Rebuttal · Authors · 2026-03-30
>
> **Thank you for the detailed review and for recognizing our core contributions. We would like to respond to each weakness as follows.**
>
> > W1: Only GPT5 for environment synthesis — generalization to other model families
>
> Thank you for your suggestion. We further conducted full pipeline synthesis (100 environments, 1,000 tasks) with Claude-4.5-Sonnet and the open-source Qwen3.5-122B-A10B. From the results below, we find that:
>
> 1. AWM is generally model-agnostic: Qwen3.5 can achieve 77% success with competitive quality.
> 2. Diversity is model-independent, confirming that our pipeline design (not the LLM) drives environmental diversity.
>
> **Pipeline Success Rate:**
> | Stage       | GPT-5 | Claude-4.5-Sonnet | Qwen3.5 |
> | ----------- | ----- | ----------------- | ------- |
> | Database    | 88.3  | 100               | 79      |
> | Sample Data | 88.2  | 100               | 97      |
> | Env Code    | 86.8  | 99                | 77      |
>
>
> **Environment Quality (evaluated by Claude-4.5-Sonnet judge):**
>
> | Quality Metric | GPT-5 | Claude-4.5-Sonnet | Qwen3.5 |
> |---|---|---|---|
> | Task Feasibility | 3.99 | 3.84 | 3.15 |
> | Data Alignment | 4.84 | 4.92 | 4.32 |
> | Toolset Completeness | 4.98 | 4.97 | 4.30 |
>
> | Bug & Task Stats | GPT-5 | Claude-4.5-Sonnet | Qwen3.5 |
> |---|---|---|---|
> | Envs w/ Bugs | 83% | 89% | 100% |
> | # Bugs per Env | 2.70 | 2.11 | 3.82 |
> | Blocked Tasks | 11.5% | 15.8% | 24.9% |
>
> **Diversity Analysis:**
>
> | Diversity Metric | GPT-5 | Claude-4.5-Sonnet | Qwen3.5 |
> |---|---|---|---|
> | Mean Pairwise Distance | 0.34 | 0.31 | 0.35 |
> | Category Coverage | 357 | 305 | 298 |
>
>
>
> > W2: OOD results not as good as claimed, especially vs EnvScaler
>
> From the results in Table 4, AWM actually performs overall better than EnvScaler on BFCLv3 and MCP-Universe, though inferior on tau2-bench.
> As briefly discussed in Sec. 5.2 (L317), EnvScaler synthesizes environments from existing task sets that may overlap with tau2-bench's specific domains (e.g., they have conversation designs fit to tau2-bench), while AWM environments are fully synthetic without targeting any benchmark. This likely explains the performance difference across benchmarks.
>
> Critically, EnvScaler's advantage on tau2-bench comes at the cost of generalization: the 8B model regresses on BFCLv3 (36.83 vs 53.83 Base) and MCP-Universe (5.59 vs 6.70 Base), while AWM improves over Base on all benchmarks.
>
> > W3: 74% environments contain bugs — need explanation
>
> We appreciate the concern. As discussed in Sec. 6.1 (L367), we report a manual inspection of 100 sampled environments with a bug taxonomy: 44% of bugs are due to not handling edge input cases (e.g., missing null/boundary validation) and 14% are operations conflicting with database constraints (e.g., violating foreign key or uniqueness). These two categories account for the majority of bugs and are typical of auto-generated code at this scale. The server error rate during our large-scale RL training is only 4% (L370), which validates that most of these bugs are non-blocking edge cases.
>
> We believe the bugs are produced due to the complexity and scale of our environments (~2,000 lines of code, 35 tools per env), and simply replacing the single LLM API call with a coding agent could help improve quality. We will make the bug taxonomy more prominent and add discussion in the limitations section.
>
>
>
> > W4: Diversity — no quantitative analysis
>
> We respectfully note that Figure 3 in our paper already provides visualized quantitative diversity analysis. Specifically:
> - (a) Embedding Diversity: we encode each environment's scenario description, DB schema, and toolset schema into embeddings and compute mean pairwise cosine distance. The distance remains stable (~0.35) as pool size grows from 10 to 1,000.
> - (b) Category Coverage: the number of unique scenario categories grows steadily to 3,619 (at 1,000), demonstrating that AWM continues to expand into new topical regions rather than producing domain-level duplicates.
>
> To further quantify diversity, we conduct a code-level similarity study on all 1,000 environments using github/mizchi/similarity (AST-based duplicate detection via Tree Structure Edit Distance) and token/n-gram Jaccard metrics.
>
> | Metric | Granularity | Mean Similarity | Max | Interpretation |
> |---|---|---|---|---|
> | AST Function Duplicate (TSED, threshold≥0.5) | function-level | **0.0%** cross-file pairs | 0.0% | Zero structurally similar functions across any pair of environments |
> | Token Jaccard | token-level | **0.183** | 0.324 | Low overlap; baseline from shared Python/FastAPI keywords |
> | Endpoint Name Jaccard | API route-level | **0.004** | — | Environments expose distinct tool interfaces |
> | Class Name Jaccard | schema-level | **0.009** | — | Database schemas are highly distinct |
>
> The results confirm the scalability of AWM's diversity: the code-level similarity is low even across 1,000 synthesized environments.

---

> > ### Author Rebuttal · Reviewer_Lzwt · 2026-04-04
> >
> > My main concerns are addressed.

---

> > > ### Author Response · Authors · 2026-04-06
> > >
> > > Thank you so much for your endorsement. We are happy to see our rebuttal has addressed your main concerns!

---

### Official Review · Reviewer_HrTs · 2026-03-13

**Soundness:** 4
**Presentation:** 3
**Significance:** 3
**Originality:** 3
**Overall Recommendation:** 5
**Confidence:** 4

**Summary:**

This paper presents Agent World Model (AWM), an end to end pipeline that automatically generates a large collection of code driven, database backed synthetic environments for training multi turn, tool using LLM agents via reinforcement learning. Instead of relying on LLM simulated environments, AWM builds executable environments with SQLite databases and MCP-exposed tools, enabling deterministic, verifiable, and scalable interaction.
The work is novel, technically solid, and empirically convincing. It is primarily a systems/methodology contribution with strong practical relevance. My overall recommendation is Strong Accept.

**Compliance With Llm Reviewing Policy:**

Affirmed.

**Final Justification:**

Final Recommendation: **Accept (5)**
Summary of Assessment
My final recommendation for this paper remains an Accept (5). The authors present a highly significant paradigm shift for training tool-using agents by moving away from LLM-simulated environments toward a robust, executable, and database-backed synthetic environment pipeline (Agent World Model). The work is technically solid, practically relevant, and introduces a scalable suite that could become a standard resource in the field.

Evaluation of Dimensions
Soundness: The methodology is rigorous and well-engineered. The transition to fully automated scenario generation, strict code-based checks, and history-aware RL training addresses real deployment mismatches. The empirical evidence of out-of-distribution (OOD) generalization is highly convincing.

Originality: The originality lies not in a single theoretical breakthrough, but in the creative integration of individual components (schema generation, RL reward shaping, LLM judgment) at an impressive scale. Shifting to an executable, deterministic environment pipeline is a highly impactful conceptual leap.

Significance: The engineering value of this work is substantial. By eliminating the need for bespoke environment construction, AWM provides immediate practical utility to researchers and practitioners spanning multiple domains (e.g., finance, e-commerce, enterprise tools).

Clarity & Presentation: The paper is systematically structured and clearly articulates the pipeline's mechanics, making the methodology highly reproducible. The distinction between prior frameworks and the proposed AWM is well-defined.

Impact of the Rebuttal
The authors provided a comprehensive and highly constructive rebuttal that effectively reinforced my prior positive assessment. My main concerns were adequately resolved:

Judge Reliability & Task Complexity: The authors supplied excellent quantitative data during the discussion phase. The high pairwise agreement metrics across LLM judges (e.g., 95.5% for GPT-5.1) and the clear stratification of performance demonstrating gains across all complexity levels successfully alleviated my concerns regarding the stability of the hybrid reward.

Concurrent Work: The clarification regarding the timeline and code unavailability of AutoForge was reasonable, and the supplementary comparison with EnvScaler effectively solidified the authors' scalability claims.

Robustness & Safety: While the authors correctly pointed out that agents encounter natural runtime errors during AWM training, the assessment of true adversarial vulnerabilities remains somewhat preliminary. However, the authors acknowledged this limitation and committed to expanding the discussion on deployment risks and adversarial conditions in the final manuscript.

Conclusion
This is a methodologically strong paper offering significant systems-level contributions to the reinforcement learning and agentic AI communities. The authors' thorough response to reviewer feedback demonstrates a commitment to rigor and transparency. I strongly encourage the authors to integrate the new tables on judge consistency and task complexity, as well as the expanded limitations discussion regarding adversarial vulnerabilities, into the camera-ready version.

**Key Questions For Authors:**

Comparison with contemporaries: Could you provide a direct empirical comparison with other synthetic environment frameworks (e.g., AutoForge)? This would clarify whether AWM is a step-change improvement or primarily a scaled variant.

Judge reliability: How consistent is the LLM judge across repeated evaluations of the same episode? Would variance in judgments affect training stability, and could alternative judges or continuous scoring improve robustness?

Task complexity: Can you stratify tasks by complexity (e.g., minimal tool call length, branching factor) and report performance by bucket? This would reveal whether gains are concentrated on simpler workflows.

Safety and robustness: How do agents trained with AWM behave under adversarial or noisy conditions (e.g., corrupted database entries, tool errors)? This is critical for assessing deployment readiness.

Theoretical discussion: Even without formal proofs, could you expand on how step-level penalties shape the optimization landscape and whether environment diversity follows a scaling law analogous to data scaling?

**Limitations:**

No. While the paper acknowledges simplified assumptions (e.g., single-node SQLite, lack of distributed systems, limited robustness evaluation), the discussion of limitations and potential negative societal impact is not sufficiently developed. Constructive suggestions include:Expanding discussion on how missing aspects (latency, failures, adversarial inputs) might affect deployment. Addressing safety concerns, such as unauthorized data access or malicious tool use.Considering societal implications of deploying RL-trained agents in sensitive domains like finance or healthcare.

**Strengths And Weaknesses:**

Soundness: The paper is technically sound and methodologically rigorous. The proposed Agent World Model (AWM) pipeline is carefully engineered, with automated scenario generation, schema synthesis, tool construction, and verification/self-correction. The reinforcement learning setup is well thought out, particularly the hybrid reward design combining strict code-based checks with LLM judgment. Empirical results are convincing, showing strong out-of-distribution (OOD) generalization and clear benefits from environment scaling and history-aware training. However, the lack of formal theoretical analysis (e.g., convergence guarantees, sample complexity bounds) and limited quantification of reward noise slightly weaken the scientific underpinning. Overall, the methods are appropriate and the claims are well supported by experiments.
Quoted evidence: “The RL component is justified purely empirically: No theoretical analysis of convergence properties under the hybrid reward.”

Presentation: The paper is clearly written, well structured, and easy to follow. The pipeline is described step by step, making the methodology reproducible. Strengths and limitations are openly acknowledged, which adds credibility. The narrative positions the work as a paradigm shift from LLM-simulated to executable environments, and this distinction is well articulated. One area for improvement is the lack of direct comparison with closely related frameworks (e.g., AutoForge), which would help contextualize novelty more precisely.

Significance: The contribution is highly significant. AWM introduces a scalable, executable environment suite that could become a standard resource for training tool-using agents. The demonstrated OOD generalization suggests practical utility beyond narrow benchmarks, potentially influencing future research directions in reinforcement learning for agents. The scope of impact is broad, spanning multiple domains (e-commerce, finance, social media, enterprise tools). The engineering value is substantial, reducing the need for bespoke environment construction and enabling reproducibility.

Originality: The work is original in its paradigm shift: moving from LLM-simulated environments to executable, database-backed synthetic environments. While the individual components (schema generation, RL reward shaping, LLM-based judgment) are not entirely novel, their integration into a fully automated pipeline is creative and impactful. The history-aware RL training regime is a particularly novel insight, addressing a real deployment mismatch. The originality lies in the combination and scale of these ideas, rather than in a single theoretical breakthrough.
Quoted evidence: “By training with the same truncated context that will be used at test time, the agent’s behaviour becomes more robust and realistic under deployment constraints.”

---

> ### Author Rebuttal · Authors · 2026-03-30
>
> **Thank you for your strong endorsement. We appreciate the detailed and constructive suggestions.**
>
> > Q1: Direct comparison with concurrent works such as AutoForge
>
> A direct comparison with AutoForge is not feasible: **AutoForge has not released any code or environments**, and appeared on arXiv Dec 28, 2025 — one month before the ICML deadline. We compare with EnvScaler (arXiv Jan 9, 2026) in Tables 3, 4, and 5. Methodology-level comparison:
>
> | Aspect | AWM | AutoForge | EnvScaler |
> |---|---|---|---|
> | # Environments | 1,000 | 10 | 191 |
> | Input requirement | Environment names only | API documentation | Task sets |
> | SQL-backed state | Yes | No | No |
> | Avg tools/env | 35.1 | N/A | 18.6 |
>
> AutoForge generates only 10 environments heavily relying on existing API documentation, which limits scalability.
>
> > Q2: LLM Judge reliability & consistency
>
> We sampled 100 agent trajectories and judged them 5 times using three LLM judges (GPT-5.1, Claude-4.5-Sonnet, Qwen3.5-122B-A10B):
>
> **Binary classification (complete vs. others) — used for RL reward:**
>
> | Metric | GPT-5.1 | Claude-4.5-Sonnet | Qwen3.5-122B |
> |---|---|---|---|
> | Self-Consistency | 90.8% | 82.0% | 76.3% |
> | Pairwise Agreement | 95.5% | 91.8% | 88.1% |
> | Fleiss' kappa | 0.891 | 0.826 | 0.728 |
> | Reward Flip Rate | 9.2% | 18.0% | 23.7% |
>
> **Detailed reward classification (complete / incomplete / server_error / agent_error):**
>
> | Metric | GPT-5.1 | Claude-4.5-Sonnet | Qwen3.5-122B |
> |---|---|---|---|
> | Self-Consistency | 82.7% | 65.0% | 69.5% |
> | Pairwise Agreement | 91.2% | 83.5% | 82.7% |
> | Fleiss' kappa | 0.781 | 0.693 | 0.650 |
> | Reward Flip Rate | 11.2% | 25.0% | 18.6% |
>
> Key findings:
> 1. Stronger reasoning models achieve higher consistency. We used GPT-5 in our RL experiments to ensure reward reliability.
> 2. Even open-source Qwen3.5 achieves 88.1% pairwise agreement, appears adequate for RL training.
> 3. Cross-model agreement validates that our verification rubrics capture genuine task completion, not model-specific artifacts.
>
> > Q3: Performance by task complexity
>
> We classify benchmark tasks by complexity and report Base vs AWM per bucket (MCP-U does not have ground truth trajectory reference or other complexity indicators). We find:
> 1. AWM improvements are NOT concentrated on simpler tasks, but across all complexity levels.
> 2. The absolute gains becomes smaller on more complex tasks, while the relative gains remain consistent, which indicates the agent's performance on hard tasks relies on its intrinsic capabilities.
>
> **BFCLv3 (8B)** — by required tool calls:
>
> | Complexity | Base Acc | AWM Acc | Delta |
> |---|---|---|---|
> | Simple (1 call) | 53.6% | 80.3% | +26.7% |
> | Parallel/Multiple (2+ calls) | 60.0% | 75.3% | +15.3% |
> | Multi-Turn (agent) | 43.9% | 45.0% | +1.1% |
>
> **tau2-bench (8B)** — by ground truth actions:
>
> | Complexity | # Tasks | Base Acc | AWM Acc | Delta |
> |---|---|---|---|---|
> | Simple (0-1 actions) | 124 | 32.7% | 41.9% | +9.3% |
> | Medium (2-4 actions) | 65 | 22.7% | 28.8% | +6.2% |
> | Hard (5+ actions) | 89 | 20.5% | 25.0% | +4.5% |
>
> Please also see Reviewer zjor W1 for task complexity analysis on AWM's synthesized tasks.
>
>
> > Q4: Adversarial/noisy conditions
>
> AWM training already exposes agents to noisy conditions: 74% of environments contain bugs (Sec. 5.1), and ~4% of RL episodes encounter environment errors at runtime. Our reward design handles this: error episodes are assigned zero rewards, preventing incorrect signals from corrupting training.
>
> Additionally, since AWM environments do not target any specific benchmark (e.g., tau-bench requires conversational tasks; BFCL has tool-rejection tests), this serves as a form of distribution shift. Agents trained on AWM show strong generalization, providing indirect evidence of robustness. We will discuss adversarial robustness as a future direction.
>
> > Q5: Theoretical discussion
>
> **Step-level penalties.** Our format penalty assigns $r_t = -1$ only to the offending step; all preceding valid steps receive $r_t = 0$. This localizes the negative signal to the offending action, making credit assignment cleaner than broadcasting a terminal penalty across the trajectory. Theoretically, this resembles potential-based reward shaping: accelerating syntactic constraint learning without altering the task-level optimal policy.
>
> **Environment scaling law.** Fig. 5 shows monotonically increasing performance from 10 to 526 environments with diminishing gains, suggestive of a power-law. However, with limited data points due to compute constraints, we cannot formally fit a scaling law yet — we leave this as a future direction.
>
> > Q6:  expanding limitations discussion
>
> Thank you for your valuable and detailed suggestions. We will expand it to discuss:
> 1. deployment risks under noisy/adversarial inputs
> 2. safety concerns around unauthorized data access/tool usage
> 3. societal implications for sensitive domains (finance, healthcare).

---

> > ### Author Rebuttal · Reviewer_HrTs · 2026-04-08
> >
> > I would like to thank the authors for their comprehensive rebuttal. The additional analyses provided during the discussion phase have clarified several of my initial questions.
> >
> > I particularly appreciate the thorough evaluation of the LLM judge's reliability (Q2). The high pairwise agreement across different models (e.g., 95.5% for GPT-5.1) and the detailed stratification of performance by task complexity (Q3) effectively alleviate my concerns regarding the stability of the hybrid reward and the precise source of the empirical gains. Furthermore, the clarification regarding the concurrent timeline and unavailability of AutoForge is perfectly reasonable, and the added comparison with EnvScaler effectively reinforces the scalability claims.
> >
> > While the rebuttal addresses most points well, the response regarding safety and robustness under adversarial conditions (Q4) remains somewhat preliminary. Arguing that the agents are robust simply because they encounter natural syntax bugs or runtime errors during AWM training is an intuitive start, but it does not fully substitute for a systematic evaluation under intentionally injected adversarial noise or maliciously corrupted database entries. This leaves the assessment of deployment readiness in high-stakes domains slightly under-explored.
> >
> >
> > Given the overall high quality of the methodology, the substantial engineering effort, and the convincing empirical results, I am happy to maintain my score of Accept (5). The paper introduces a highly significant paradigm shift for training tool-using agents.If the paper is accepted, I strongly encourage the authors to integrate the new tables on judge consistency and task complexity into the final version, and to ensure the promised expanded discussion on limitations, particularly regarding adversarial vulnerabilities, is fully incorporated.

---

> > > ### Author Response · Authors · 2026-04-08
> > >
> > > Thank you for your thoughtful acknowledgement and for maintaining your positive assessment of our work. We are glad that our rebuttal helped clarify your concerns, especially regarding judge reliability, task complexity, and comparison to concurrent work.
> > >
> > > We really appreciate your constructive note on safety and robustness under adversarial conditions. Your comment highlights an important issue: natural syntax bugs or runtime errors during AWM training may not comprehensively cover adversarial scenarios, and targeted malicious attacks are a valuable scenario to discuss.  In the final version, we will incorporate these new analyses results and expand the discussion of limitations and deployment risks especially regarding such adversarial vulnerabilities.
> > >
> > > Thank you again for your time and effort in reviewing our work!

---

### Official Review · Reviewer_zjor · 2026-03-13

**Soundness:** 3
**Presentation:** 3
**Significance:** 3
**Originality:** 3
**Overall Recommendation:** 4
**Confidence:** 2

**Summary:**

This work proposes a pipeline for generating synthetic environments for LLM agent training. Each environment is built by first generating a scenario description and some concrete user tasks that function as the environment’s requirements. Given these tasks, it synthesizes a minimal DB schema and corresponding seed data such that every task is executable from the initial state. It then generates a tool/API specification together with a Python MCP server. Finally, the method synthesizes task-specific verification code over pre/post database states and combines these programmatic signals with an LLM judge to assess whether a task has been completed. The author uses the synthesized environments for online GRPO, with a hybrid reward that penalizes wrong-format tool calls and uses a code-augmented LLM judge, grounded in database-state verification, to score whether the task is solved. Experiments show that training on these synthesized environments improves OOD tool-use performance.

**Compliance With Llm Reviewing Policy:**

Affirmed.

**Final Justification:**

Main concerns have been well addressed.

**Key Questions For Authors:**

see Weaknesses

**Limitations:**

yes

**Strengths And Weaknesses:**

**strengths**:

1. The paper proposes a comprehensive pipeline for synthesizing tool-use environments for post-training LLM agents.

2. The synthesized environments could serve as a useful resource for future work.

3. The experimental results show that LLM agents can benefit from training in the environments synthesized by the proposed method.

**weaknesses**:

1. it's unclear how to synthesize more challenging task using the proposed pipeline. Relatedly, it is uncertain whether the LLM can write reliable verification code if the synthesized task is very complicated.

2. While the authors claim that their synthesized dataset is of higher quality than prior work, they also generate substantially more environments than previous methods e.g. EnvScaler. Therefore, it is not rigorous to attribute the improvement entirely to data quality rather than data quantity.

3. The author name their proposed data synthesize pipeline Agent world model. The use of the term world model feels somewhat odd here, given that it already has a well-established meaning in the ML and RL literature

---

> ### Author Rebuttal · Authors · 2026-03-30
>
> **Thank you for recognizing the comprehensive pipeline design and the value of our synthesized environments. We would like to answer your questions by point-to-point responses.**
>
> > W1: Unclear how to synthesize more challenging tasks / reliable verification for complex tasks
>
> We agree this is an important question. We address it from two perspectives: (1) demonstrating that current AWM tasks are already challenging, and (2) outlining how to scale complexity further.
>
> **Are AWM tasks challenging enough?** We present a complexity-stratified analysis of all 10,000 synthesized tasks, evaluated by GPT-5.1 and Claude-4.5-Sonnet, two of the most powerful frontier models. Overall, they achieve only 36.1% and 62.6% Pass@1 respectively, confirming that AWM tasks are non-trivial. (GPT-5.1's lower rate may be partly due to its routing strategy and tendency to refuse following the system prompt.)
>
> | Complexity Bucket | Tasks Ratio | Avg Steps | GPT-5.1 Pass@1 | Sonnet 4.5 Pass@1 | Both Pass | Neither Pass |
> |---|---|---|---|---|---|---|
> | Simple (1-3 calls) | 35.1% | 3.3 | 61.7% | 68.4% | 45.3% | 15.2% |
> | Medium (4-6 calls) | 31.5% | 5.8 | 27.0% | 71.8% | 24.9% | 26.2% |
> | Hard (7-10 calls) | 15.7% | 9.1 | 11.9% | 63.3% | 11.5% | 36.3% |
> | Very Hard (11+ calls) | 17.7% | 17.9 | 3.0% | 31.0% | 3.0% | 69.0% |
>
> Pass rates decrease monotonically with complexity for both models. For "Very Hard" tasks, **69% are unsolved by either model**. Furthermore, from our RL training logs, 27.1%, 23.7%, and 28.2% of tasks are never solved even once by Qwen-4B, 8B, and 14B throughout the entire training process (14B only optimizes for 32 steps), providing further evidence that AWM generates genuinely challenging tasks that remain difficult even after extensive RL exploration.
>
> **How to further scale complexity with reliable verification?** After environment synthesis, we can construct new tasks by sampling from existing database records and tool endpoints with additional preconditions and constraints, naturally increasing complexity. Rejection sampling can filter out trivially easy or infeasibly hard tasks. Since environments are fully transparent to the verifier, the same verification pipeline applies without modification. Generating multiple independent verification paths per task and ensembling results can further improve reliability. We plan to include this in the future work discussion.
>
> > W2: Quality vs. quantity — cannot attribute improvement entirely to quality
>
> Thank you for your suggestion. Actually, we have already conducted environment scaling experiments in Sec. 6.4 that directly address this question.
> The scaling curve shows that AWM with only 100 environments already outperforms EnvScaler (191 environments) on both BFCL and MCP-Universe.
> This indicates that AWM's improvement is not solely driven by quantity: even with fewer environments, the quality advantage is evident.
>
> To further ensure a fair comparison, we trained AWM with the exact same number of environments as EnvScaler.
> Due to limited time and compute budget, we ran only 32 optimization steps.
> Despite this reduced training, AWM with 191 environments still outperforms EnvScaler on two benchmarks.
>
> | Method | # Envs | BFCL | Tau | MCP |
> |---|---|---|---|---|
> | EnvScaler | 191 | 54.06 | 28.96 | 4.47 |
> | AWM | 10 | 48.05 | 14.21 | 3.91 |
> | AWM | 100 | 57.01 | 16.55 | 6.14 |
> | AWM (32 step) | 191 | 57.56 | 16.37 | 6.14 |
>
> For tau-bench, AWM underperforms EnvScaler because EnvScaler synthesizes environments from existing task sets that may overlap with tau-bench's specific domains, while AWM environments are fully synthetic without targeting any benchmark. Please refer to Sec. 5.2 for a detailed discussion.
>
> Moreover, we believe the ability to scale to a large number of diverse environments is itself a major contribution of our synthesis pipeline.
> This scalability demonstrates the potential of AWM as a general-purpose environment factory for agent training.
>
>
>
> > W3: "World Model" naming feels odd
>
> We appreciate this feedback. Our naming aims to reflect the functional role: AWM serves as a "world model" for agent training by providing executable state transitions: the agent interacts with AWM environments to learn about how the world responds to its actions, analogous to the role of learned world models in model-based RL. The key difference is implementation: AWM uses code-driven environments rather than learned neural dynamics. We would like to add a clarification in the introduction to avoid confusion.

---

> > ### Author Rebuttal · Reviewer_zjor · 2026-04-04
> >
> > Thank you for the supplementary results. I am leaning toward accepting the paper.

---

> > > ### Author Response · Authors · 2026-04-06
> > >
> > > Thank you so much for your endorsement. We are happy to see our rebuttal has addressed your concerns!

---

### Decision · Program_Chairs · 2026-04-30

**Decision:**

Accept (regular)

**Comment:**

The paper proposes Agent World Model (AWM), a fully synthetic pipeline for generating code-driven, database-backed environments for training multi-turn tool-use LLM agents via RL. All four reviewers converged on acceptance (scores 4, 4, 5, 5), with all concerns marked fully resolved. Reviewers highlighted the pipeline's practical value and scalability (1,000 environments, 35 tools/env on average), the convincing OOD generalization results, and the thorough quality/diversity analyses.

The rebuttal strengthened the paper with additional evidence: cross-model pipeline validation (Claude, Qwen), LLM judge consistency metrics, task complexity stratification showing gains across all difficulty levels, and quantitative code-level diversity analysis.

The authors should incorporate these supplementary results and expand the limitations discussion on adversarial robustness in the camera-ready version.